# An efficient urine peptidomics workflow identifies chemically defined dietary gluten peptides from patients with celiac disease

Brad A. Palanski[1,13,14], Nielson Weng[1,2,3,14], Lichao Zhang [4], Andrew J. Hilmer[1], Lalla A. Fall[5], Kavya Swaminathan [6], Bana Jabri[7,8,9], Carolina Sousa [10], Nielsen Q. Fernandez-Becker[11], Chaitan Khosla[1,5,12,15✉] & Joshua E. Elias [4,15✉]

Celiac disease (CeD) is an autoimmune disorder induced by consuming gluten proteins from wheat, barley, and rye. Glutens resist gastrointestinal proteolysis, resulting in peptides that elicit inflammation in patients with CeD. Despite well-established connections between glutens and CeD, chemically defined, bioavailable peptides produced from dietary proteins have never been identified from humans in an unbiased manner. This is largely attributable to technical challenges, impeding our knowledge of potentially diverse peptide species that encounter the immune system. Here, we develop a liquid chromatographic-mass spectrometric workflow for untargeted sequence analysis of the urinary peptidome. We detect over 600 distinct dietary peptides, of which ~35% have a CeD-relevant T cell epitope and ~5% are known to stimulate innate immune responses. Remarkably, gluten peptides from patients with CeD qualitatively and quantitatively differ from controls. Our results provide a new foundation for understanding gluten immunogenicity, improving CeD management, and characterizing the dietary and urinary peptidomes.

[1] Department of Chemistry, Stanford University, Stanford, CA, USA. [2] School of Medicine, Stanford University, Stanford, CA, USA. [3] Medical Scientist Training Program, Stanford University, Stanford, CA, USA. [4] Chan Zuckerberg Biohub, San Francisco, CA, USA. [5] Stanford ChEM-H, Stanford University, Stanford, CA, USA. [6] Division of Blood and Bone Marrow Transplantation, Stanford University, Stanford, CA, USA. [7] Department of Medicine, University of Chicago, Chicago, IL, USA. [8] Committee on Immunology, University of Chicago, Chicago, IL, USA. [9] Department of Pathology and Pediatrics, University of Chicago, Chicago, IL, USA. [10] Facultad de Farmacia, Departamento de Microbiología y Parasitología, Universidad de Sevilla, Sevilla, Spain. [11] Division of Gastroenterology and Hepatology, Department of Medicine, Stanford University, Stanford, CA, USA. [12] Department of Chemical Engineering, Stanford University, Stanford, CA, USA. [13] Present address: Department of Medicine, Brigham and Women's Hospital, and Department of Biological Chemistry and Molecular Pharmacology, Harvard Medical School, Boston, MA, USA. [14] These authors contributed equally: Brad A. Palanski, Nielson Weng. [15] These authors jointly supervised this work: Chaitan Khosla, Joshua E. Elias. ✉email: khosla@stanford.edu; josh.elias@czbiohub.org

In humans, the prevailing understanding of physiological digestion is that proteins are broken down into single amino acids or di- or tri-peptides before absorption as nutrients[1,2]. Gluten proteins found in wheat, barley, and rye are exceptions to this tenet, which results in notable health consequences. Unusual biochemical properties of these proteins, such as a high abundance of glutamine (Gln, Q) and proline (Pro, P) residues, render them resistant to degradation by gastrointestinal proteases[3]. Consequently, relatively long gluten peptides with intact immunotoxic epitopes accumulate in the lumen of the small intestine[4] and cross the epithelial barrier, although the mechanism(s) of transport remain controversial[5]. In approximately 1 in 100 individuals, an aberrant immune response to these peptides causes celiac disease (CeD), an autoimmune disorder that causes small intestinal mucosal injury characterized by villous atrophy. Common symptoms include abdominal pain, bloating, nausea, vomiting, and/or diarrhea[6]. Extraintestinal CeD manifestations also occur, including blistering skin rashes and ataxia[7]. At present, the only effective CeD treatment is a strict, lifelong adherence to a gluten-free diet (GFD)[8].

Assays to probe gluten immunogenicity in CeD have typically relied on extracted or recombinant glutens that are digested in vitro with gastrointestinal proteases. The resulting peptides can be used in bioassays to characterize mechanisms of immunotoxicity[3]. In most cases, these peptides elicit adaptive immune responses mediated by HLA-DQ2 or -DQ8 antigen presentation to Th1 cells in the small intestinal mucosa of patients with CeD[4,9–14]. Gluten peptides with alternative (i.e., non-T cell-dependent) modes of action have also been reported[15–21]. Structure-function analyses using synthetic gluten peptides have revealed exquisite sequence specificity for both the HLA and T-cell receptor; single amino acid alterations can result in dramatically altered affinities, thereby altering the strength of the immune response[22–24].

Although it is well established that chemical variation in gluten peptides can influence CeD immune responses, measuring their naturally processed forms and connecting them with CeD patients' health status is an underexplored research area. Over 20 years ago, chromatographic analysis coupled to UV detection implied the existence of gluten-derived peptides in the urine of patients with CeD[25]. This was confirmed more recently by antibody-based methods[26–29]. Indeed, most current gluten detection methods rely on monoclonal antibodies, which recognize amino acid motifs present in a subset of gluten proteins[30]. Notwithstanding the valuable knowledge that has been gained from analyzing biospecimens with these immunoreagents, they are neither capable of revealing the exact gluten peptide sequences that are present, nor are they comprehensive in that some CeD-relevant peptides may lack the epitopes these antibodies recognize.

To address mounting evidence that many disease-relevant gluten peptides remain to be discovered[31], new methods are needed to recover and precisely characterize in vivo gluten digestion products in an untargeted fashion. In fact, not a single chemically defined peptide from wheat (or, to our knowledge, from any dietary protein) has ever been identified from the human circulatory or excretory systems. Consequently, very little is known about the physiological absorption, distribution, metabolism, and excretion (ADME) of gluten. Specifically, we lack insight into the chemical structures of peptides formed from in vivo digestion that may stimulate the immune system.

As a step toward filling this knowledge gap, we sought to analyze human urine by liquid chromatography coupled to tandem mass spectrometry (LC–MS/MS). Currently, LC–MS/MS is the most widely used technique for peptide sequencing in complex biological samples[32]. However, established LC–MS/MS methods suffer from technical limitations when applied to urinary peptidome analysis. High concentrations of urinary salts and metabolites, which are not efficiently removed by standard reversed-phase or liquid-liquid extraction procedures, can overwhelm chromatography systems and interfere with peptide detection[33,34]. Here, we develop a sample preparation and LC–MS/MS method that utilizes mixed cation exchange solid-phase extraction to exclude these interfering molecules in a single step. This workflow overcomes problems we initially encountered with adapting established methods for urinary peptidomics, such as the need for time-consuming strong cation exchange purification and/or limited depth of peptide sampling. With it, we can now efficiently identify dietary gluten peptides and report the precise sequences of such peptides in the urine of human volunteers. We also undertake an exploratory clinical study, which revealed wheat-derived peptides that are substantially different in their chemical and biological properties and are differentially found in patients with CeD versus healthy controls. These peptides are attractive candidates for improving CeD diagnosis and for monitoring patient compliance to GFDs. They also set the stage for elucidating mechanisms underlying the anomalous ADME characteristics of gluten and other dietary proteins. More generally, the successful application of our urinary peptidomic workflow to CeD suggests it should be broadly applicable for the direct measurement of both endogenous and exogenous peptides present in urine.

## Results

**LC–MS/MS method development.** Previous chromatographic and antibody-based detection methods suggested the presence of gluten-derived peptides in human urine but were incapable of directly elucidating their sequences[25–29]. To determine if an unbiased LC–MS/MS approach could be used for this purpose, we initially analyzed urine samples from volunteers who had consumed a meal rich in dietary gluten by following an extraction and analysis protocol typically used in LC–MS/MS-based proteomics studies (Supplementary Methods). These exploratory efforts yielded just one gluten-derived peptide identification. Furthermore, very few endogenous human peptides were detected (Supplementary Fig. 1). In accordance with challenges previously reported with urinary peptidome experiments[33,34], we noted severe degradation of the LC column and MS electrospray source over consecutive analyses, preventing us from analyzing multiple samples without unacceptable interruptions in instrument operation.

We, therefore, endeavored to systematically optimize each step in our peptidome extraction and LC–MS/MS analytical protocols in order to reliably detect peptides from urine. Starting with urine from an individual challenged with dietary gluten, we first confirmed it had measurable gluten levels by ELISA. The commercially available R5 monoclonal antibody we used targets the pentameric motif QXP(W/F)P found in many gluten proteins[35] and is therefore expected to react with gluten peptides produced by patients with CeD and healthy controls. Using this reference specimen, we tested a variety of methods to enrich and purify urinary peptides. In addition, we optimized the LC gradient and MS acquisition method (Supplementary Methods). These efforts resulted in 29- and 13-fold increases in the number of identifiable wheat- and human-derived peptides, respectively, while eliminating longstanding problems with instrument contamination by background metabolites (Supplementary Methods, Supplementary Figs. 2 and 3, see also Method Specificity section below). Importantly, by combining solid-phase extraction on a mixed cation exchange column with online reversed-phase LC and high-resolution MS/MS

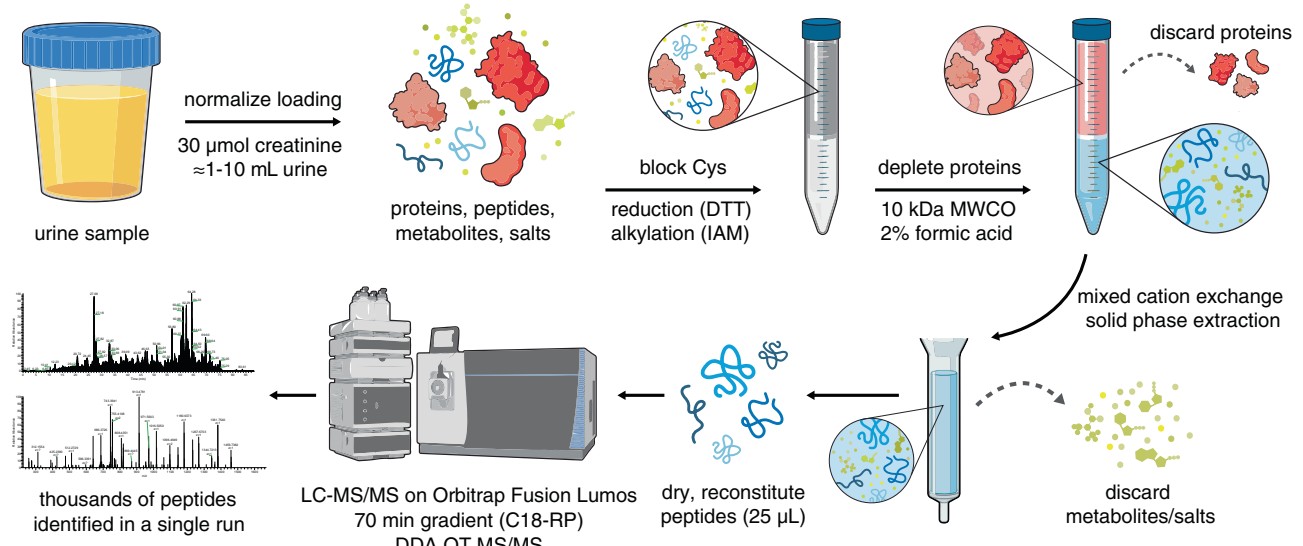

**Fig. 1 An efficient chemical extraction and LC–MS/MS workflow for urinary peptidomic analysis.** Urine sample volumes are first normalized based on creatinine measurement. To prevent the formation of intra- or intermolecular disulfide bonds that would confound downstream data analysis, cysteine (Cys) residues are reduced with dithiothreitol (DTT) and alkylated with iodoacetamide (IAM). Proteins are depleted using a centrifugal filtration device with a molecular weight cutoff (MWCO) of 10 kDa under acidic denaturing conditions. The filtrate is processed using a mixed cation exchange solid-phase extraction column to remove metabolites and salts that interfere with LC–MS/MS analysis. After extensive washing, peptides are eluted from the column with methanol containing 5% ammonium hydroxide. Eluted peptides are dried in a centrifugal vacuum concentrator and reconstituted in water. Peptides are separated by nano-liquid chromatography (LC) on a reversed-phase column (RP-C18) and analyzed by tandem mass spectrometry (MS/MS) on an Orbitrap Fusion Lumos mass spectrometer operated in the data-dependent acquisition (DDA) mode, with precursor and fragment ions analyzed in high resolution in the Orbitrap (OT). The resulting LC–MS/MS data are searched against the desired databases (e.g., the human and wheat proteomes). The experimental workflow takes ~6 h to complete and results in the identification of thousands of peptides.

analysis, this workflow shortened our sample preparation time from 2–3 days to 6 h (Fig. 1). This allowed sufficient throughput for us to undertake a comparative analysis of clinical urine samples.

**Pilot identification of chemically defined urinary wheat peptides.** To evaluate our optimized urine peptidome extraction and LC–MS/MS method, four healthy participants were recruited into a pilot study outlined as follows (Fig. 2a): On the first day, participants initiated a GFD. The GFD was maintained on the second day, and a pooled urine sample was collected over 8 h. On the third day, participants underwent a gluten challenge comprised of two wheat bagels (~18 g gluten), and subsequently collected a second pooled urine specimen over the next 8 h. LC–MS/MS analysis of these urine samples yielded an average of 24 unique peptide sequences per participant mapping to the wheat proteome, all of which were identified only in the post-gluten challenge samples (Supplementary Dataset 3). The number of unique wheat peptides varied widely between individual volunteers (Fig. 2b). In contrast, the number of detected human peptides was essentially unchanged in the GFD versus post-challenge samples (Supplementary Fig. 4).

In anticipation of recruiting larger cohorts of human participants, we sought to confirm our initial findings while also simplifying the dietary gluten challenge protocol. The four original participants, along with four additional healthy participants, were asked to fast overnight in place of the 48 h GFD employed in the pilot study (Fig. 2c). Then, a single urine sample was collected before consumption of two wheat bagels for breakfast and pooled urine collection over the next 8 h. Wheat peptides were readily detected in all post-challenge samples (Fig. 2d and Supplementary Dataset 4), while the number of human peptides was similar in both sample groups (Supplementary Fig. 5). These results support our initial finding that a dietary gluten challenge consistently leads to urinary gluten peptides that are measurable by LC–MS/MS. They also indicate that levels of these peptides fall to undetectable levels after overnight fasting, justifying the use of the abbreviated protocol for subsequent studies.

Considering all sequences of the wheat-derived peptides identified from healthy participants (Supplementary Datasets 3 and 4), four noteworthy characteristics were apparent (Table 1). First, although 123 unique peptide sequences were identified, only one peptide was found in common in all 8 participants: GQQQPFPPQQPYPQPQPFPS. Its C-terminally truncated derivative, GQQQPFPPQQPYPQPQPFP, was identified in 7/8 participants; other overlapping variants of these peptides were also detected in more than one urine sample. While these sequences from α-gliadin proteins are not known to be recognized by patient-derived T cells, multiple studies have demonstrated that they can stimulate an innate immune response[15–21]. Strikingly, these prevalent urinary peptides do not have a motif that is recognized by monoclonal antibodies commonly used for gluten detection[35,36]. Thus, they would evade detection by traditional immunoassays.

Second, we identified many peptides harboring known CeD-relevant T-cell epitopes[37]. The DQ2.5-glia-γ4c/DQ8-glia-γ1a epitope was most common, with 32 distinct peptides harboring the corresponding QQPQQPFPQ sequence. The DQ2.5-glia-γ5 epitope (QQPFPQQPQ) also appeared frequently, as it was identified in 14 unique peptides. Other known T-cell epitopes, including DQ2.5-glia-γ1/DQ8.5-glia-γ1/DQ8-glia-γ2 (PQQSFPQQQ), DQ2.5-glia-γ3/DQ8-glia-γ1b (QQPQQPYPQ), and DQ2.5-glut-L1/DQ2.2-glut-L1 (PFSQQQQPV) were identified less frequently (Supplementary Table 1). Some T-cell stimulatory peptides, such as FLQPQQPFPQQPQQPYPQQPQQPFPQ and SQQPQQPFPQPQQQFPQPQQPQ, harbor more than one epitope and were previously identified as being both proteolytically resistant and

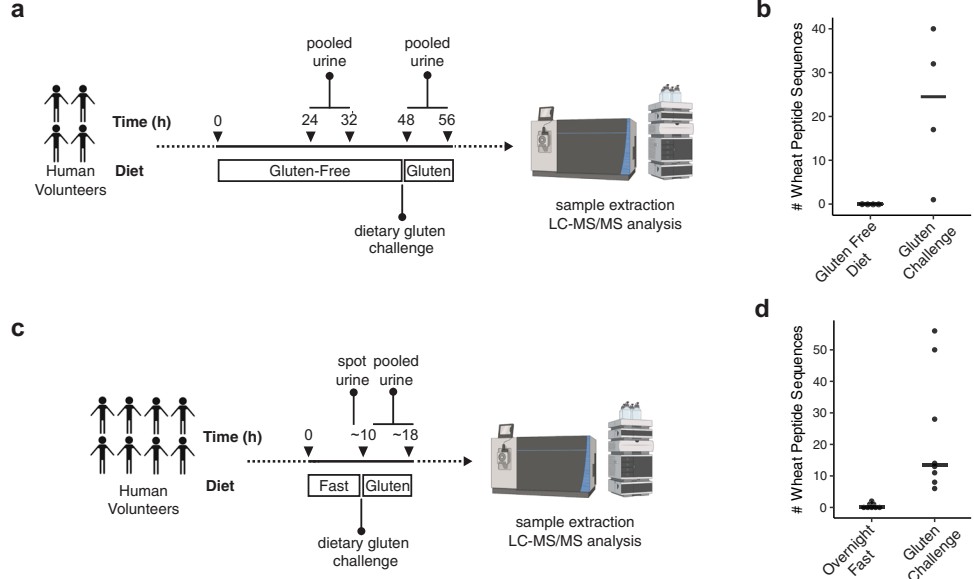

**Fig. 2 LC–MS/MS enables the identification of chemically defined dietary wheat peptides in human urine. a** Pilot study design. **b** Number of wheat-derived peptide sequences detected in the urine of 4 healthy participants (4 males, 24-28 years of age) after a 24-h gluten-free diet and after a dietary challenge with wheat gluten. Horizontal line represents median. **c** Simplified study design. **d** Number of wheat-derived peptide sequences detected in 8 healthy participants (5 males, 3 females, 24-28 years of age) after an overnight fast and after a dietary challenge with wheat gluten. Horizontal line represents median. In all LC–MS/MS experiments, peptide identifications were controlled at a false discovery rate of 1% using the PEAKS software decoy-fusion approach[59]. Source data are provided as a Source Data file.

highly inflammatory to CeD patients through in silico and in vitro analysis[38].

Third, some peptides such as PyrQTFPHQPQQQVPQPQQ PQQP had undergone post-translational modification via cyclization of an N-terminal Gln into a pyroglutamic acid residue. The possibility that this modification occurs during sample processing cannot be excluded. However, N-terminal pyroglutamation is known to protect peptides from proteolytic degradation[39–41] and thus may contribute to their stability in circulation.

Last, we found gluten peptides from human urine that have not been described by any prior study as either being resistant to gastrointestinal digestion or as having any pathophysiological characteristics. One such peptide, SCHVMQQQCC, is derived from γ-gliadin and from the low-molecular-weight subunits of glutenins B and C. It was identified in 6/8 participants, while its C-terminally extended form was detected in 5/8 participants. Longer and shorter versions were also detected in multiple samples. This result illustrates the power our untargeted approach has for identifying wheat-derived peptides that have eluded prior description despite their seemingly common representation.

**Method specificity**. To our knowledge, untargeted detection of diet-derived peptides in humans had not been reported prior to our proof-of-concept studies above. We, therefore, sought to confirm that our LC–MS/MS method reliably detects these peptides and accurately identifies their sequences in the complex background of the human urinary peptidome. To do so, we took advantage of the fact that rye and barley also contain proteolytically resistant gluten proteins[42]. For example, wheat gliadins (one of two major protein families that comprise gluten) are homologous to secalin proteins in rye and hordeins in barley (Fig. 3a). We thus anticipated that urine analysis should distinguish the particular grain consumed by volunteers who ate meals rich in either wheat, barley, or rye. To test this hypothesis, two healthy participants fasted overnight, and then, on different days, maintained a grain-free diet, or ate breakfasts prepared with

approximately 1.5 cups of wheat, barley, or rye flour. Urine was then collected for 8 h, analyzed by our optimized LC–MS/MS protocol, and the data were searched against a customized database that included the human, rye, wheat, and barley proteomes (see "Methods" section).

When participants maintained a grain-free diet, only a single grain peptide (likely a false discovery) was identified. In contrast, subsequent to wheat, rye, and barley dietary challenges, we respectively detected 51, 43, and 37 unique grain peptide sequences (Supplementary Dataset 5). The human peptide repertoire was similar in all specimens (Supplementary Fig. 6). Notably, all identified peptides were restricted to just one dietary challenge (Fig. 3b), suggesting that our method readily differentiates peptides formed from different dietary proteins. Indeed, when the sequences were mapped onto the grain proteomes, the majority were predominately derived from the corresponding dietary challenge (Fig. 3c and Supplementary Fig. 7). Our observation that fewer sequences could be mapped to the rye proteome from rye challenge urine is a likely artifact of the size of the available rye proteome: the UniProt resource contains ~100× fewer sequences than the wheat or barley proteomes, despite their genomes having similar sizes. Our data suggest that many of the detected sequences in the rye challenge urine may be derived from yet unannotated rye proteins whose sequences are also present in the wheat proteome. Taken together, these results demonstrate that the peptides detected by our method originate from the diet. They also confirm the specificity of our method to distinguish closely related dietary components.

**Preliminary analysis of banked urine from patients with CeD**. To test our ability to identify wheat-derived peptides in the urine of individuals with CeD, we analyzed a set of nine urine samples banked from a prior study[27]. These urine samples were collected from single voids of patients with confirmed CeD diagnoses and were previously assayed for gluten presence by lateral flow immunoassay. This immunoassay used G12 and A1 monoclonal

**Table 1 Examples of wheat-derived peptide sequences identified in human urine.**

| Wheat peptide sequence[a] | Uniprot protein accession(s)[b] | Protein name(s) | Mass (Da) | # of participants with peptide |
|---|---|---|---|---|
| GQQQPFPPQQPYPQPQPFPS | GDA0_WHEAT; GDA1_WHEAT; GDA2_WHEAT; GDA9_WHEAT | α/β-gliadin; α-gliadin | 2292.0962 | 8 |
| GQQQPFPPQQPYPQPQPFP | GDA0_WHEAT; GDA1_WHEAT; GDA2_WHEAT; GDA9_WHEAT | α/β-gliadin; α-gliadin | 2205.0642 | 7 |
| SQQPEQTIS**QQPQQPFPQ**QPHQPQQPYPQQQPYGSSL | A0A290XYU3_WHEAT; U5UA46_WHEAT; R9XT67_WHEAT; R9XUY1_WHEAT | ω-gliadin | 4284.0259 | 6 |
| SCHVMQQQCC | GDB1_WHEAT; GLTC_WHEAT; GDB3_WHEAT; GLTB_WHEAT | γ-gliadin; LMW glutenin | 1336.4781 | 6 |
| LGQQQPFPPQQPYPQPQPFPSQQP | GDA1_WHEAT; A0A0K2QJX_WHEAT; A0A0E3Z7F7_WHEAT; A0A1K0JNE4_WHEAT | α/β-gliadin; α-gliadin | 2758.3500 | 5 |
| **SQQPQQPFPQ**QPHQPQQPYPQ | B6ETR9_WHEAT; A0A290XYU3_WHEAT; U5UA46_WHEAT; R9XT67_WHEAT | ω-gliadin; LMW glutenin | 2512.1882 | 5 |
| SCHVMQQQCCQ | GDB1_WHEAT; GLTC_WHEAT; GDB3_WHEAT; GLTB_WHEAT | γ-gliadin; LMW glutenin | 1464.5367 | 5 |
| PyrQTFPHQPQQQVPQPQQQQP | GDB2_WHEAT; B6DQB5_WHEAT; R9XWD0_WHEAT; R9XUB9_WHEAT | γ-gliadin | 2348.1296 | 5 |
| PQQPFSQQQQQQQQQQQQ**PPFSQQQQPVL** | Q5MFQ2_WHEAT; D0EVP4_WHEAT; Q5MFQ1_WHEAT; Q5MFQ6_WHEAT; R9XUS6_WHEAT; A0A290XZ20_WHEAT; A0A290XZ51_WHEAT; A0A290XZ34_WHEAT | LMW glutenin | 3457.6763 | 3 |
| **TQQPQQPFPQQPQ**QPFP**QQPQQPFPQ** | GDB2_WHEAT; B6DQB5_WHEAT, R9XV87_WHEAT; R9XWD0_WHEAT | γ-gliadin | 3098.4998 | 3 |
| **TQQPQQPFPQQPQ**QPFPQT**QQPQQPFPQ** | GDB2_WHEAT; B6DQB5_WHEAT; R9XV87_WHEAT; R9XWD0_WHEAT | γ-gliadin | 3327.6060 | 3 |
| QPFPQQPYPQPQPFP | GDA0_WHEAT; GDA1_WHEAT; GDA2_WHEAT; GDA9_WHEAT | α/β-gliadin; α-gliadin | 1891.9257 | 2 |
| P**[I/L]QPQQPFPQQPQQPFPQ**PQ[c] | B6ETR9_WHEAT; A0A290XYW6_WHEAT; R9XT67_WHEAT; R9XUY1_WHEAT | ω-gliadin | 2354.1807 | 2 |
| **FLQPQQPFPQQPQQPYPQQPQQPFPQ** | GDB0_WHEAT, R9XV78_WHEAT; Q9FS77_WHEAT, B6UKM8_WHEAT | γ-gliadin | 3145.5410 | 1 |
| CHVMQQQCCQ[d] | GDB1_WHEAT; GLTC_WHEAT; GDB3_WHEAT; GLTB_WHEAT | γ-gliadin; LMW glutenin | 1393.5000 | 1 |
| PyrQQQQPPFSQQPPISQQQQPPFSQQQQPQF | GLTA_WHEAT, D2D1I3_WHEAT, Q6SPZ1_WHEAT, Q6SPZ3_WHEAT | LMW glutenin | 3416.6174 | 1 |

[a]Residues in bold indicate CeD-relevant T-cell epitopes, as defined in ref. [37]. All Cys residues were detected in their carbamidomethylated forms due to derivatization with iodoacetamide during sample workup. "PyrQ" indicates glutamine residues that had undergone cyclization into pyroglutamic acid.
[b]For peptides mapping to more than four proteins in the UniProt database, only the first four accession codes are displayed. All identified peptide sequences and full details relating to peptide identification in each participants' urine are provided in Supplementary Datasets 3 and 4.
[c]The UniProt database contained otherwise identical peptides containing Leu and Ile at the same position. These two amino acids are isobaric and indistinguishable by our MS/MS method and thus the sequence is reported as [I/L].
[d]The Met in this peptide was detected in its oxidized form.

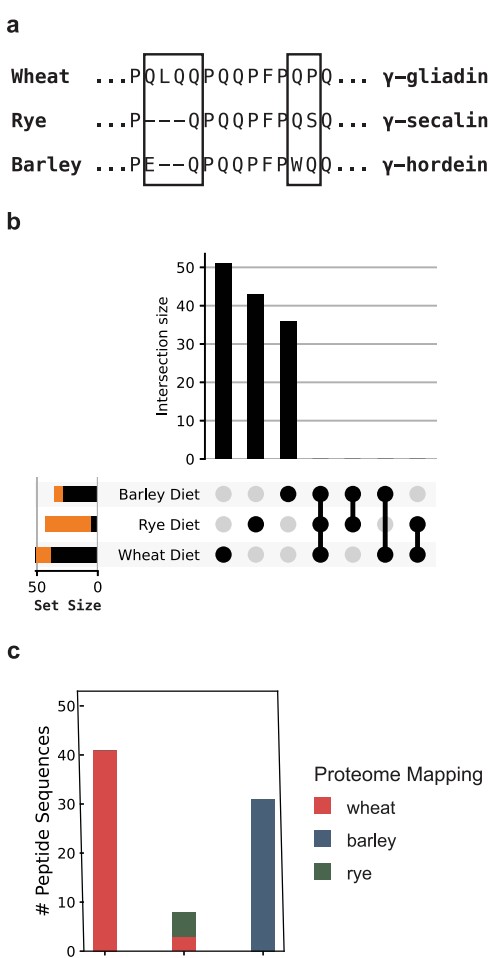

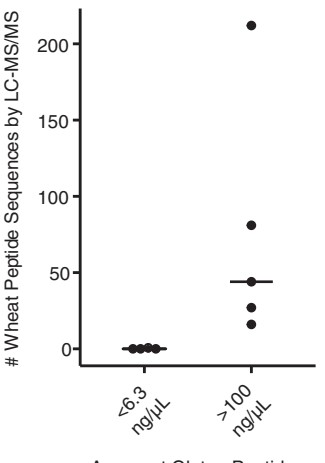

**Fig. 4 LC-MS/MS analysis of banked urine samples from patients with CeD[27].** Urine collected from a single void was tested for the presence of gluten by an A1 and G12 monoclonal antibody-based lateral flow assay. Each data point represents a single urine sample, and the horizontal bar indicates the median. Information associated with the identification of these peptide sequences by LC–MS/MS is provided in Supplementary Dataset 6. Source data are provided as a Source Data file.

In contrast, all samples ($n = 5$) for which lateral flow assays indicated high apparent gluten peptide concentrations, yielded at least 17 wheat peptide sequences (Fig. 4) by LC–MS/MS. The mean number of identified wheat peptides (74 per sample) was much higher than those described in our proof-of-concept studies with healthy participants (24 per sample; Fig. 2b, d) while the number of human peptides was similar (Supplementary Fig. 8). Strikingly, one sample from a patient with CeD yielded 206 distinct wheat peptides, a quantity four times greater than in any healthy participant analyzed previously. These intriguing observations led us to hypothesize that the urinary wheat peptide repertoires of patients with CeD are much more diverse than non-CeD individuals. We, therefore, initiated a prospective clinical study to formally test this hypothesis, described below.

**Clinical study**. Adult patients with gastrointestinal symptoms including dyspepsia, bloating and diarrhea who were eating gluten-containing diets and were undergoing evaluation for CeD were recruited within the Celiac Disease Program at the Stanford Digestive Health Center, along with gender-matched healthy controls (Fig. 5a). CeD vs. non-CeD diagnoses were not known at the time of enrollment and were subsequently established by serology and endoscopy with biopsy. All participants undertook a dietary gluten challenge and urine collection according to the optimized study design depicted in Fig. 2c. At the end of the recruitment period, urine samples were processed and analyzed by LC–MS/MS. Compared to healthy controls ($n = 8$) and patients ultimately diagnosed with non-celiac gastrointestinal disorders ($n = 5$), patients with CeD ($n = 6$) had approximately 5 times more unique gluten peptides in their urine (Fig. 5b), while the number of endogenous human peptides was similar in all three groups (Supplementary Fig. 9).

Consistent with our pilot studies, variants of the innate immune response stimulating-peptide GQQQPFPPQQPYPQPQPFPS were detected in all individuals, regardless of clinical status (Supplementary Dataset 7). We analyzed whether patients with CeD had increased diversity of peptides with T-cell epitopes known to stimulate the adaptive immune response. Indeed, the number of peptides with at least one CeD-relevant T-cell epitope was

**Fig. 3 LC-MS/MS analysis of urine from two healthy participants challenged with dietary wheat, rye, and barley. a** Illustration of sequence differences between closely related gliadin, secalin, and hordein proteins from wheat, rye, and barley, respectively. The boxes highlight source-specific sequence regions. **b** All possible intersections of the three sets of peptide sequences identified after the grain challenges are depicted as an UpSet plot[60]. The individual set size is plotted to the left of each row, with the black portion of the bar representing sequences that map uniquely to the respective dietary challenge proteome, and the orange portion representing those that do not. There are no intersections between these sets of peptides, suggesting the peptides depicted in orange are nonetheless likely to have originated from the specific grain used for the dietary challenge. **c** Number of distinct peptide sequences uniquely mapping to the wheat, barley, and rye proteomes following dietary challenges. Sequences that mapped to more than one of the proteomes were excluded from this figure. Sequences that mapped to more than one of the proteomes were excluded from this figure. In the rye diet, sequences mapping uniquely to the wheat proteome were only found in the rye diet, suggesting they are yet unannotated rye peptides. A full list of observed peptide sequences and their respective proteome mappings is provided in Supplementary Dataset 5. Participants in this experiment were males, ages 27 and 28. Source data are provided as a Source Data file.

antibodies, which specifically recognize the amino acid motifs QPQLP(Y/F) and QLP(Y/F)PQP, respectively[36]. Samples ($n = 4$) from patients reporting adherence to a GFD and in which the measured gluten peptide concentrations were below the lateral flow assay limit of quantification were analyzed as negative controls. In accordance with the lateral flow results, we did not measure peptides mapping to the wheat proteome by LC–MS/MS.

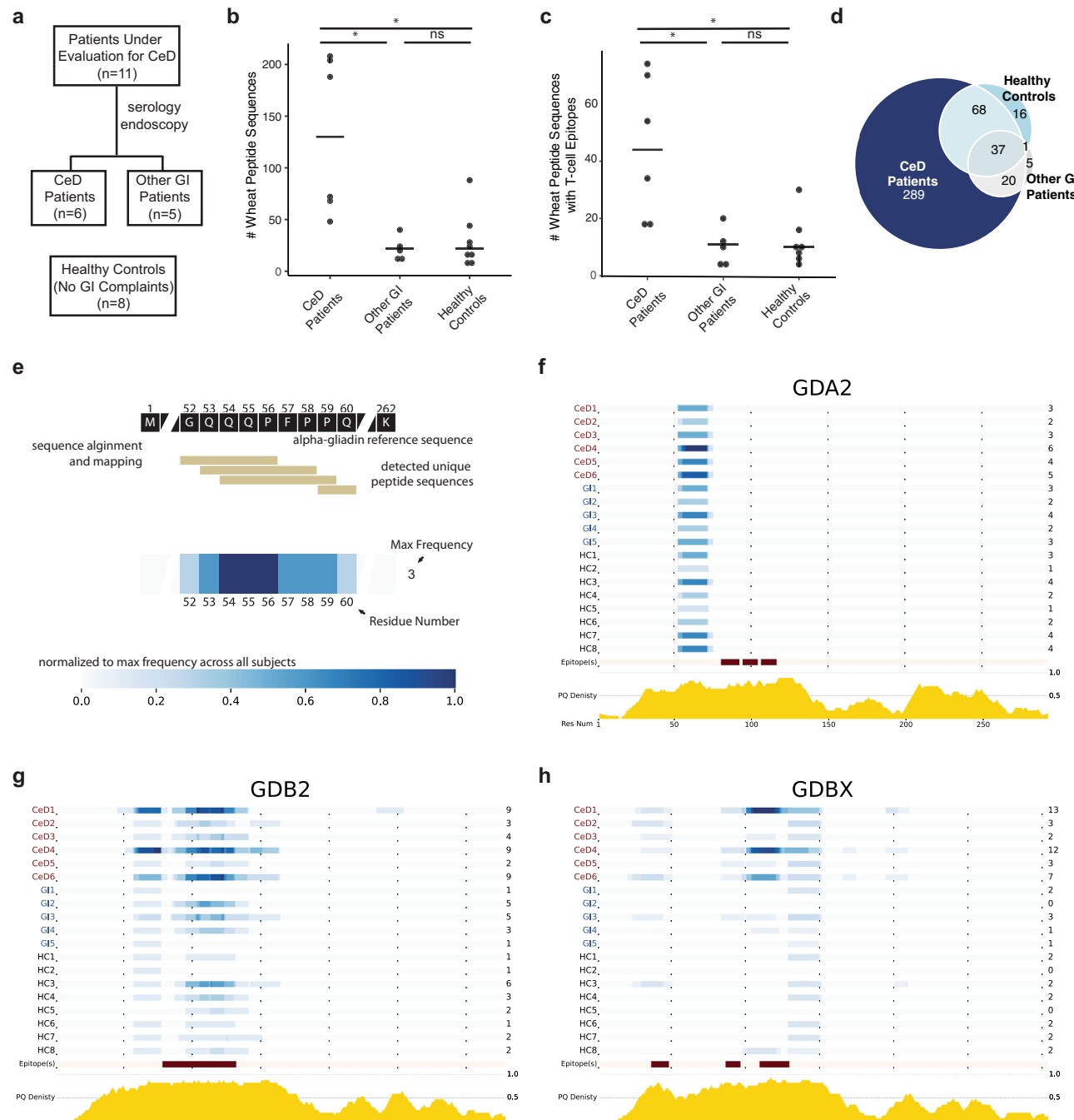

approximately 4-fold higher in patients with CeD (Fig. 5c). CeD-relevant T-cell epitopes occurred more frequently in participants with CeD, and seven different T-cell epitopes completely absent in controls were found in patients (Table 2 and Supplementary Table 2). Overall, we measured 289 unique wheat peptides from patients with CeD that were not detected in healthy individuals or patients with non-celiac gastrointestinal disorders, whereas only 37 peptides were shared between all three groups (Fig. 5d and Supplementary Dataset 7).

To gain insight into the diversity of peptide sequences found in patients with CeD, we ranked the frequency by which CeD-specific peptide sequences were detected (Supplementary Dataset 8). Just four peptides were found in the majority of (≥4/6) patients (Table 3), whereas most (201) were detected only in single individuals. To understand the chemical space occupied by the peptides that were found in patients with CeD, we aligned the

detected peptide sequences to the wheat proteome (Fig. 5e). Strikingly, patients with CeD not only had more variants (i.e., slightly longer or shorter versions) of peptides found in controls, but they also possessed peptides that mapped to distinct regions of the wheat proteome (Fig. 5f–h and Supplementary Fig. 10). For example, three patients had peptides spanning residues 194–199 of γ-gliadin (GDBX) that were absent in controls. Taken together, these results demonstrate that patients with CeD have peptidomes that are more diverse in their chemical and biological properties compared to control participants.

## Discussion

Despite extensive research into the mechanisms by which gluten proteins play a role in CeD pathogenesis (Fig. 6), the identities of the actual diet-derived molecules that interact with the human

**Fig. 5 The urinary wheat peptidomes of patients with CeD are significantly more diverse than in healthy controls or patients with non-celiac gastrointestinal disorders. a** Clinical study design. Participants were recruited over approximately 2 years within the Celiac Disease Program at the Stanford Digestive Health Center. Participant characteristics are reported in Supplementary Tables 3 and 4. **b** Number of wheat peptides detected in pooled urine samples collected for 8 h subsequent to a dietary challenge with two bagels (~18 g gluten). Patients with CeD had substantially more wheat peptide sequences compared to healthy controls ($p = 0.017$) or patients with non-celiac gastrointestinal disorders ($p = 0.017$). **c** Number of unique wheat-derived peptide sequences with at least one T-cell epitope in patients with CeD compared to healthy controls ($p = 0.019$) and patients with non-celiac gastrointestinal disorders ($p = 0.046$). **b**, **c** Statistics were derived from $n = 6$ patients with celiac disease, $n = 5$ patients with non-celiac gastrointestinal disorders, and $n = 8$ healthy controls using a one-way Kruskal–Wallis ANOVA/Dunn's multiple comparison test; horizontal bar represents the median. **d** Venn diagram comparing the unique peptide sequences detected in healthy controls, patients with CeD, and other GI patients. A full listing of peptide sequences is provided in Supplementary Dataset 7. **e** Schematic describing how the frequencies at which urinary wheat-derived peptides map to proteins in the wheat proteome were plotted as heatmaps. **f–h** Heatmap representation of detected peptide sequences in alpha/beta gliadin 2 (GDA2), γ-gliadin 2 (GDB2), and gamma-gliadin X (GDBX) demonstrates that while many peptides map to the same region of the wheat proteome, the urinary wheat peptidomes of patients with CeD also occupy distinct chemical space. Below the maps, the locations of CeD T-cell epitopes are highlighted in red, and the relative density of Gln and Pro is indicated in yellow. Heatmaps for other proteins in the wheat proteome are provided in Supplementary Fig. 10. In **b–h**, all samples were analyzed by LC–MS/MS in duplicate and the aggregated results are shown. Analyses of individual replicates are provided in Supplementary Fig. 11. Full details on LC–MS/MS identification of peptide sequences are provided in Supplementary Dataset 7. Source data are provided as a Source Data file.

**Table 2 Number of detected T-cell epitopes in patients with celiac disease (CeD), non-celiac gastrointestinal disorders (GI), and healthy controls (HC).**

| Epitope name(s)[a] | Epitope sequence | CeD | GI | HC |
|---|---|---|---|---|
| DQ2.5-glia-α1a | PFPQPQLPY | 2 | 0 | 0 |
| DQ2.5-glia-α1b | PYPQPQLPY | 1 | 0 | 0 |
| DQ2.5-glia-α2 | PQPQLPYPQ | 2 | 0 | 0 |
| DQ2.5-glia-α3 | FRPQQPYPQ | 1 | 0 | 0 |
| DQ2.5-glia-γ1/DQ8.5-glia-γ1/DQ8-glia-γ2 | PQQSFPQQQ | 1 | 0 | 1 |
| DQ2.5-glia-γ3/DQ8-glia-γ1b | QQPQQPYPQ | 6 | 1 | 0 |
| DQ2.5-glia-γ4a | SQPQQQFPQ | 1 | 0 | 0 |
| DQ2.5-glia-γ4b | PQPQQQFPQ | 8 | 4 | 3 |
| DQ2.5-glia-γ4c/DQ8-glia-γ1a | QQPQQPFPQ | 105 | 29 | 36 |
| DQ2.5-glia-γ4d | PQPQQPFCQ | 2 | 0 | 1 |
| DQ2.5-glia-γ4e | LQPQQPFPQ | 28 | 5 | 4 |
| DQ2.5-glia-γ5 | QQPFPQQPQ | 37 | 10 | 15 |
| DQ2.5-glia-ω1/DQ2.5-hor-1/DQ2.5-sec-1 | PFPQPQQPF | 3 | 0 | 1 |
| DQ2.5-glia-ω2 | PQPQQPFPW | 1 | 0 | 0 |
| DQ2.5-glut-L1/DQ2.2-glut-L1 | PFSQQQPV | 17 | 2 | 11 |
| DQ2.5-glut-L2 | FSQQQQSPF | 1 | 0 | 0 |
| DQ8.5-glut-H1/DQ8-glut-H1 | QGYYPTSPQ | 2 | 0 | 1 |

[a]Epitope sequences and nomenclature are defined in ref. [37].

salts that are difficult to separate from peptides and are incompatible with LC–MS/MS[33,34]. Therefore, we developed an extraction technique to remove these interfering compounds. This allowed us to achieve our main goal of wheat peptide identification, while also improving our ability to measure urinary peptides originating from other endogenous and dietary sources (Supplementary Fig. 3). Our workflow (Fig. 1) is compatible with standard reversed-phase LC–MS/MS instrumentation available in most proteomics laboratories[32]. Compared to published approaches employing solid-phase extraction techniques, our method identifies approximately 2 to 10 times more endogenous human peptides from typical 1–10 mL urine samples[33,34,43,44]. Moreover, our sample preparation technique requires only ~6 h, facilitating sufficient throughput for us to undertake a comparative analysis of the wheat-derived urinary peptidomes of patients with CeD and healthy controls. Given its high specificity (Fig. 3), our workflow has potential utility for other studies requiring analysis of the urinary peptidome. Although here we focused on the identification of wheat-derived peptides, we also identified over 30,000 human peptides (Supplementary Datasets 1–7), which is to our knowledge, is the largest collection of urinary peptides sequenced by LC–MS/MS to date. Parenthetically, we note that previous surveys of the urine peptidome found that hydroxyproline-modified collagen peptides were among the most abundant[33,34,43,44]. However, here we did not allow hydroxyproline as a variable modification in our database searches, as this modification was not relevant to our overriding goal of detecting biologically relevant, wheat-derived peptides. Therefore, we have deposited our raw data in the PRIDE database to facilitate the identification of additional peptides (e.g., by including other variable modifications or using alternative search engines).

This study has identified, for the first time, the specific amino acid sequences and post-translational modifications of peptides resulting from in vivo digestion of gluten. In our pilot identification of dietary wheat peptides in healthy participants (Fig. 2), we identified sequences with known CeD relevance, as well as others that had never been implicated in CeD or identified as being resistant to gastrointestinal digestion. These findings underscored the value of our untargeted LC–MS/MS method and motivated us to analyze the urine of patients with CeD.

Using banked urine specimens, we found that wheat-consuming patients with CeD had substantially greater gluten peptide diversity than non-CeD individuals (Figs. 2 and 4). We confirmed this finding with a subsequent controlled clinical study (Fig. 5) in which participants followed our well-defined gluten challenge protocol (Fig. 2d). Mechanistically, this peptide

immune system in the gut and other organs have been a long-standing mystery. As a step toward illuminating these molecular interactions, we developed a LC–MS/MS-based assay that directly detects the metabolic products of food grains. In contrast to in vitro gluten preparations that have been standard research tools in the CeD field for several decades, our method is compatible with in vivo dietary challenge. It therefore can reveal the precise chemical structures of dietary peptides that may drive CeD in humans. Furthermore, unlike traditional antibody reagents, the untargeted LC–MS/MS readout readily distinguishes the precise amino acid sequences of digestion-resistant peptides from wheat, barley, and rye. Thus, this method provides a unique window into gluten's ADME characteristics and immunogenicity. Indeed, we observed more varied wheat peptide repertoires from patients with CeD than was previously appreciated – and ones that were much more diverse than those found in non-CeD individuals.

Initially, our efforts were hindered by LC–MS/MS methods that were unsuited to the unique challenges of urinary peptidome analysis. Urine contains high concentrations of metabolites and

**Table 3 Wheat-derived peptide sequences detected in the majority of patients with CeD but not in control groups.**

| Wheat peptide sequence | Protein name | # of CeD patients with peptide |
|---|---|---|
| PQQQIPQQHQIPQQPQQFPQQ | ω-gliadin | 5 |
| GQQQQQFPGQQQPFPPQQPYPQPQP | α/β-gliadin | 4 |
| PELQQPIPQQPQQPFPLQPQQPFPQQ | ω-gliadin | 4 |
| PTSPQQSGQGQQPGQWQQPGQGQPG | HMW-glutenin | 4 |

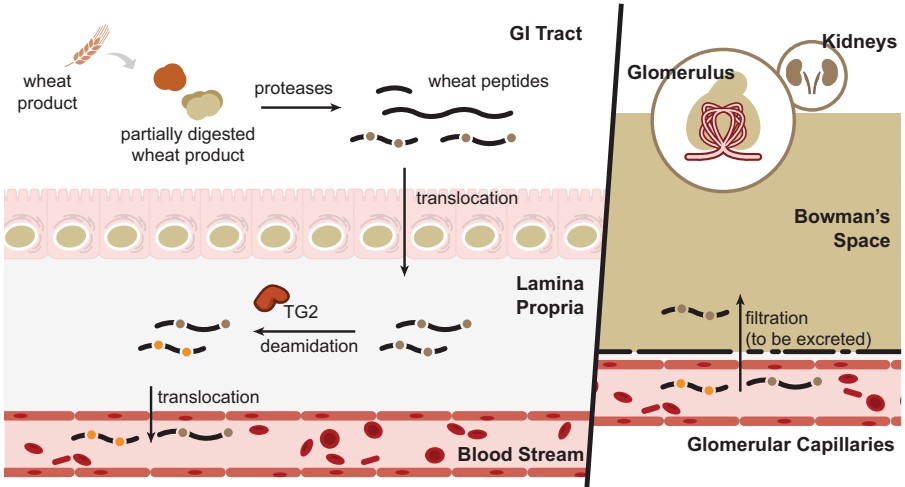

**Fig. 6 Schematic representation of relevant metabolic products of dietary gluten in humans.** Wheat-derived proteins such as gluten are resistant to gastrointestinal proteolysis. Incomplete digestion preserves disease-causing T-cell epitopes in peptides that accumulate in the gut lumen. A portion of these gluten peptides translocate to the lamina propria where transglutaminase 2 (TG2) deamidates selected Gln residues (brown dots) in some sequences, thereby converting them into Glu residues (yellow dots). Absorbed peptides can also enter the bloodstream, and are further metabolized and/ or excreted in urine.

diversity may stem from the fact that individuals with active CeD have elevated intestinal permeability, which likely allows wheat-derived peptides to more readily cross the epithelial barrier[5]. This phenomenon could lead to increased peptide concentrations in the systemic circulation and in urine, thereby enhancing their ability to be detected. A different (or complementary) explanation is that patients with CeD digest and/or metabolize wheat differently than other individuals. Such potential differences in wheat metabolism could contribute to the onset of CeD. Future analyses of patients with other non-CeD enteropathies known to cause "leaky gut" such as tropical sprue as well as disorders in which the mucosa remains grossly intact such as irritable bowel syndrome with diarrhea should be insightful in pinpointing the origins of the increased peptide diversity observed in patients with CeD[45,46]. More fundamentally, analysis of urine from individuals consuming diets with various food additives, such as microbial transglutaminase, should help to address the emerging hypothesis that these additives contain increased levels of immunogenic peptides and/or directly contribute to increases in intestinal permeability[47,48].

We identified 17 distinct immunotoxic epitopes known to be presented on MHC class II molecules and recognized by T cells that reside in the lamina propria of patients with CeD. Many of these epitopes occurred more frequently in patients with CeD than in controls (Table 2). Notably, however, we were unable to detect in its intact form an extensively studied and highly inflammatory 33-residue peptide from α2-gliadin that accumulates in the mammalian gut lumen under physiological conditions by virtue of its resistance to gastrointestinal proteolysis[4]. Its apparent absence from human urine could be due to physico-chemical properties that hinder absorption, promote systemic

metabolism, or make it unamenable to chemical extraction and/or detection by mass spectrometry. Nonetheless, it is just as noteworthy that an analogous 26-residue peptide from γ-gliadin, FLQPQQPFPQQPQQPYPQQPQQPFPQ, was observed in 2/7 patients with CeD. This peptide is also proteolytically resistant and highly inflammatory to CeD-specific T cells on account of its polyvalency[38].

Despite the abundance of immunogenic wheat peptides in urine, thus far we have been unable to definitively identify any peptide that underwent regioselective Gln deamidation by transglutaminase 2 (TG2), an important step in CeD pathogenesis (Fig. 6). Because TG2-catalyzed deamidation increases the immunogenicity of gluten-derived peptides by several orders of magnitude[4], it is possible that a relatively small molar fraction of absorbed gluten is deamidated in patients with CeD, making the concentrations of these peptide species too low to be detectable. Alternatively (or additionally), deamidation may enhance the metabolic lability of circulating gluten peptides. The possibility that their high affinity for HLA-DQ2 or -DQ8 alters their ADME characteristics should not be overlooked. Regardless, future development of targeted LC–MS/MS methods could facilitate the detection of low-abundance deamidated gluten peptides.

There is increasing interest in non-invasive monitoring of GFD compliance via urinalysis[26–28,49], as a large proportion of patients with CeD experience symptoms even when attempting to adhere to a GFD[50]. Available methods predominantly exploit the G12 monoclonal antibody, which preferentially recognizes the immunotoxic QPQLP(Y/F) motif found in several CeD-relevant T-cell epitopes[36]. Our data suggest that this motif is rare in urine from individuals who consume dietary wheat (Table 2). In contrast, the sequence GQQQPFPPQQPYPQPQPFPS appeared in

the urine of every gluten-challenged individual in this study regardless of CeD status. Most urine samples also contained longer or shorter variants of this sequence. Interestingly, a variant of this peptide has been widely investigated for its ability to actuate the innate immune response by stimulating the production of IL-15[17–21], a key cytokine in CeD pathogenesis[51,52]. At a minimum, our findings suggest that antibody-based detection methods for monitoring GFD compliance could be vastly improved by targeting detection of this common urinary peptide.

The results reported here pave the way for designing improved diagnostic methods for CeD. Although we observed a statistically significant increase in the number of wheat-derived peptides in patients with CeD compared to healthy controls and individuals with other gastrointestinal aliments, the range of unique peptide sequences was quite broad, suggesting that peptide number alone is unlikely to be sufficiently sensitive or selective for CeD diagnosis. Only a small subset (~2%) of 289 CeD-unique peptides occurred in the majority of patients, while ~75% of these sequences were uniquely detected in single individuals (Supplementary Dataset 8). This broad distribution of detected peptide sequences likely stems from the intrinsically high variability of protein digestion[53], as well as the fact that CeD is a highly heterogenous disease in terms of symptoms and extent of gastrointestinal damage[54]. Nonetheless, a subset of diet-derived peptide sequences tended to occur more frequently in CeD patients (Fig. 5f–h and Supplementary Dataset 8). Undoubtedly, future studies using larger sample cohorts will be required to identify and validate strong candidate peptides for diagnostic purposes. Such validation studies should benefit from the development of targeted LC–MS/MS methods that reproducibly sample and quantify the peptides of interest.

Beyond monitoring and diagnosing CeD, our urine peptidomic workflow has the potential to advance our fundamental understanding of gluten immunogenicity. Recent evidence indicates that only half of gluten-reactive T cells recognize a known epitope[31]. Analysis of urine from gluten-challenged individuals with T cells of unknown sequence specificity may reveal candidate epitopes that can be tested in immunoassays. Knowledge of the complete repertoire of disease-relevant wheat peptide sequences that encounter the immune system in vivo will be critical to advance efforts to treat CeD by sequestering, degrading, or blocking the interaction of these peptides with immune cells[8].

Generally, our sample extraction and LC–MS/MS analysis protocol promises to reveal new biological insights currently hidden in the urinary peptidome. For example, there is a long-standing interest in bioactive peptides derived from dietary proteins other than gluten, but little direct evidence for their existence in humans[2]. More broadly, while it is appreciated that urinary peptides may reflect an individual's health status, few peptides have been identified as disease biomarkers[55]. The high specificity and throughput afforded by our urinary peptidomics workflow should enable investigations into questions such as these that require identification of the sequences and posttranslational modifications of exogenous or endogenous urinary peptides.

## Methods

**Human urine donors**. Control urine samples were obtained from healthy adult volunteers without any gastrointestinal symptoms suggestive of CeD. Urine samples were also obtained from eleven adult participants undergoing evaluation for suspected CeD within the Celiac Disease Program at the Stanford Digestive Health Center, in accordance with a protocol approved by the Stanford institutional review board (#20362). All participants provided informed consent. Subsequent to urine collection, six of these individuals were diagnosed with CeD on the basis of abnormally high anti-TG2 and deamidated gliadin peptide (DGP) antibody titers and a confirmatory endoscopy with biopsies demonstrating villous atrophy, in accordance with currently accepted diagnostic guidelines[56]. Five individuals were

determined to have non-celiac gastrointestinal ailments on the basis of negative anti-TG2 and DGP antibody titers, and optically and histologically normal small bowel on endoscopy (with the exception of one patient who refused endoscopy). All participants provided informed consent, and ethical approval for these sample collections was obtained from the Institutional Review Board for Human Subject Research of Stanford University (Stanford, CA). Urine samples from participants analyzed in Fig. 5 were frozen on the day of collection and stored until the end of the 2-year study recruitment period. All samples were then defrosted and processed on the same day, as detailed below. Further details related to these participants are provided in Supplementary Table 3. The data in Fig. 4 were obtained from analysis of banked urine samples of individuals with a confirmed CeD diagnosis from a previously published study[27]. Participants provided informed consent, and the local ethics committee of the Hospital Universitario Virgen del Rocío (Sevilla, Spain) approved the study.

**Urine conditioning and immunochromatographic test for detection of gluten immunogenic peptides**. Urine samples were processed according to the manufacturer's recommendations (iVYCHECK GIP Urine, Biomedal S.L., Seville, Spain), and subsequent to the processing of the sample: 100 µL of the sample was added onto the detection test strip. After 30 min, the immunochromatographic strip was measured in the cassette of the lateral flow test reader, essentially as previously described[29].

**Gluten challenge and urine collection**. In initial experiments, participants followed a gluten-free diet for one day, and then continued the gluten-free diet and collected a pooled urine sample for 8 h on the second day, beginning with the second morning void. On the third day, volunteers underwent a dietary gluten challenge consisting of two bagels made with wheat flour (approximately 9 g gluten each). After the gluten challenge, participants were permitted to eat ad libitum, and a pooled urine sample was collected for 8 h. For studies that required a barley or rye enriched diet, pancakes prepared with approximately 1.5 cups of barley (Arrowhead Mills), rye (Bob's Red Mill), or wheat (Bob's Red Mill) flour were substituted for bagels prepared with wheat flour, and participants refrained from eating any wheat, rye, or barley products until the end of the 8 h urine collection. In later experiments, volunteers were not prescribed a gluten-free diet the day prior to the gluten challenge but were instructed to fast overnight. After fasting, participants collected a spot urine sample, and underwent the dietary gluten challenge as previously described. Subsequent to this gluten challenge, a pooled urine sample was collected for 8 h in a 4 L urine container (Simport Scientific #B3504L), whereas spot urine samples were collected in 120 mL urine collection cups (Thermo Scientific #010001). Urine was stored at 4 °C throughout the collection day and was preprocessed and frozen on the same day it was collected. To do so, urine samples were centrifuged in 50-mL polypropylene tubes for 3000 × g for 20 min and transferred to 15 mL polypropylene tubes in 10–15 mL aliquots. Samples were stored at −80 °C until further analysis.

**Creatinine normalization and urinary peptidome enrichment**. At the time of analysis, urine samples were placed in a 37 °C water bath until just thawed and then centrifuged at 5000 × g for 5 min at room temperature to pellet any precipitates. For all analyses, urine samples were processed in singlicate, except for the clinical study presented in Fig. 5, in which samples were processed in duplicate on two separate days. Creatinine levels were measured using a kit (Cayman Chemical #500701) according to the manufacturer protocol. Briefly, urine sample aliquots were diluted 1:10 in MilliQ water and 15 µL of the diluted samples, or 15 µL of the kit-provided creatinine standard (0–20 mg/dL final concentration, also diluted in water) were added to a 96-well plate followed by 150 µL of alkaline picrate solution. After incubation for 10 min at room temperature, the initial absorbances at 500 nm were determined on a plate reader. The reaction was quenched with 5 µL of the kit-provided acid solution, incubated for 20 min, and the final absorbances at 500 nm were measured. Final absorbance values were subtracted from the initial values, and a calibration curve using the creatinine standards was constructed. The creatinine concentrations from the urine samples were calculated based on this curve. If a urine sample reading fell out of the linear range, the measurement was repeated using an appropriate dilution. Then, a volume of urine containing ca. 30 µmol creatinine (1–10 mL for most donors) was neutralized by the addition of aqueous 1 M ammonium bicarbonate solution to a final concentration of 50 mM. Samples were reduced for 30 min at room temperature by dithiothreitol (500 mM aqueous stock added to a final concentration of 2 mM) and then alkylated in the dark with iodoacetamide (500 mM aqueous stock added to a final concentration of 4 mM) for 30 min at room temperature. Urine was acidified with mass spectrometry-grade formic acid to a final concentration of 2% (v/v). To enrich the low-molecular-weight urinary peptidome, the acidified samples were applied to water-washed Vivaspin Centrifugal Filtration Columns (Sartorius, #VS0602) with a molecular weight cutoff of 10 kDa, and centrifuged at 4000 × g until less than 200 µL urine remained in the retentate. The filtrate was then processed by solid-phase extraction, as described below.

**Solid-phase extraction of urinary peptides**. To remove salts, metabolites, and urinary pigments that interfere with downstream LC–MS/MS analysis, solid-phase

extraction of peptidome-enriched urine was performed using Oasis Mixed Cation Exchange Prime Columns (3 cc size, 100 mg resin, Waters Corporation #186008918). The filtrate from the prior centrifugal filtration step was applied at a rate of approximately 2 drops/s. The column was washed with 1 mL of an aqueous solution of 2% formic acid containing 100 mM ammonium formate, followed by 1 mL methanol. The resin was re-equilibrated with 2 mL aqueous 2% formic acid, then washed with 1 mL 95% water/5% ammonium hydroxide followed by 1 mL 95% acetonitrile/5% ammonium hydroxide. The reequilibration and wash steps were repeated one additional time. Peptides were eluted in 1 mL 95% methanol/5% ammonium hydroxide directly into a low-binding microcentrifuge tube (Fisher #3453) at a rate of approximately 1 drop every 2 s and dried using vacuum centrifugation. Dried peptides were stored at 4 °C.

**LC-MS/MS sample analysis**. At the time of analysis, solid-phase extracted urine samples were reconstituted in 25 μL MilliQ water. The resuspended peptides were centrifuged at $16,300 \times g$ for 10 min, and the supernatants were transferred to low-binding ultra-performance liquid chromatography vials (Wheaton #11-0000-100-S) and analyzed on a Fusion Lumos mass spectrometer (ThermoFisher Scientific, San Jose, USA). Peptides were separated by capillary reverse-phase chromatography on a 24 cm reversed-phase column (100 μm inner diameter, packed in-house with ReproSil-Pur C18-AQ 3.0 μm resin from Dr. Maisch GmbH). The Fusion Lumos was equipped with a Dionex Ultimate 3000 LC system and used a two-step linear gradient with 4–25% buffer B (0.1% (v/v) formic acid in acetonitrile) for 50 min followed by 25–50% buffer B for 20 min, where buffer A was 0.1% (v/v) formic acid in water. Sample analysis with the Fusion Lumos system (Tune 3.3) was carried out in top speed data-dependent mode with a duty cycle time of 3 s. Full MS scans were acquired in the Orbitrap mass analyzer with a resolution of 120,000 (FWHM) and $m/z$ scan range of 400–1500. Precursor ions with charge state 2–7 and intensity threshold above 50,000 were selected for fragmentation using collision-induced dissociation (CID) with quadrupole isolation, isolation window of 1.6 m/z, normalized collision energy of 35%, activation time of 10 ms, and activation Q of 0.25. Precursor ions with charge state 3–7 were also selected for fragmentation using electron transfer dissociation (ETD) supplemented with 25% collision energy (EThcD). Calibrated charge-dependent ETD parameters were enabled. MS2 fragment ions were analyzed in the Orbitrap mass analyzer with a resolution of 15,000 (FWHM) and $m/z$ scan range of 156–2000. Fragmented precursor ions were dynamically excluded from further selection for a period of 30 s. The AGC target was set to 400,000 and 50,000 for full FTMS scans and FTMS2 scans. The maximum injection time was set to "auto" for full FTMS scans and to "dynamic" for FTMS2 scans.

**LC-MS/MS data analysis**. To identify peptides originating from the human and wheat proteomes, and in some cases, the barley and rye proteomes, the corresponding protein sequences were downloaded from UniProt (www.uniprot.org) to generate custom FASTA databases, as detailed below. The raw MS data were searched against these databases using PEAKS software (versions X or X+, Bioinformatics Solutions, Inc.). Carbamidomethylation (on Cys) was set as a fixed modification and deamidation (on Asn and Gln), oxidation (on Met), and N-terminal pyroglutamination (on Gln and Glu) were allowed as variable modifications, with a maximum of 3 variable modifications per peptide. The allowed mass tolerances were 10 ppm for precursor ions and 0.02 Da for product ions. The digest mode was "unspecific". Peptides identified in each urine sample were filtered to a false discovery rate of 1% using the decoy-fusion approach implemented in PEAKS. Sequences were assigned to the human and wheat proteomes, and wheat peptide assignments were manually validated to ensure the correct assignment of deamidation, as detailed below. Figures were generated using R version 3.6.1 and Python version 3.6.8, and GraphPad Prism version 9.

**FASTA database construction for identification of wheat-derived peptides**. The protein database used to search for peptides from the human and wheat proteomes (data in Figs. 1, 3, and 4), was curated from the UniProt repository (www.uniprot.com) as follows. Protein sequences from the SwissProt databases (containing manually curated protein sequences) of the human (*Homo sapiens*) and wheat (*Triticum aestivium*) taxonomies were concatenated into a single FASTA file. Additionally, all protein sequences from the wheat taxonomy that contained the terms "gliadin", or "glutenin", in the TrEMBL database (containing automatically curated sequences from genome sequencing) were also downloaded and added to the FASTA file containing the SwissProt sequences. All sequences were downloaded from UniProt on December 7, 2019, and the FASTA file has been deposited in the PRIDE repository (https://www.ebi.ac.uk/pride/) under accession number PXD031048.

**FASTA database construction for identification of wheat, rye, and barley derived peptides**. The protein database used to search for peptides from the human, wheat, barley, and rye proteomes (data in Fig. 3) was curated from the UniProt repository as follows. Protein sequences from the SwissProt databases of the human (*Homo sapiens*), wheat (*Triticum aestivium*), rye (*Secale cereale*), and barley (*Hordeum vulgare*) taxonomies were concatenated into a single FASTA file. Additionally, all protein sequences from the wheat, rye, and barley taxonomies that

contained the terms "gliadin", "glutenin", "secalin", or "hordein" in the TrEMBL databases were also downloaded and added to the FASTA file containing the SwissProt sequences. All sequences were downloaded from UniProt on October 7, 2020, and the FASTA file has been deposited in the PRIDE repository (https://www.ebi.ac.uk/pride/) under accession number PXD031048.

**Assignment of peptide sequences to the human and wheat proteomes**. Inspection of the PEAKS output indicated that the software truncated, omitted, or otherwise incorrectly parsed the protein accession codes for some peptides (possibly due to the large size of the grain databases that were searched). A Python script was therefore used to generate the "Accession(s)" column reported in the supplementary datasets, ensuring that all protein accession codes corresponding to a given peptide are reported. For downstream data analysis, peptide sequences were assigned to the wheat proteome only if they uniquely mapped to the wheat proteome; sequences that mapped to both the wheat and human proteomes were counted as human peptides. Additionally, peptides mapping to the wheat proteome containing the subsequence YVRPD were assigned as human peptides even if the complete peptide sequence was uniquely mapped to the wheat proteome. In a preliminary analysis of our proof-of-concept studies presented in Fig. 2, we observed that peptides containing this subsequence appeared regardless of gluten dietary status. BLAST searching indicated that this subsequence appears in the immunoglobulin heavy chain joining region. We speculate that the complete peptide sequences were formed as the result of V-(D)-J rearrangement of human immunoglobin genes[57].

**Validation of wheat-derived peptide sequences identified by PEAKS software**. The output of the PEAKS software used to identify peptide sequence from the LC–MS/MS data suggested that some wheat peptides were present in their deamidated forms. The mass difference between a deamidated peptide and the second isotopic peak of a native peptide (containing one $^{13}$C) is only 0.0193 Da. Thus, aberrant deamidation assignments commonly occur in proteomics studies when the second isotopic peak of a native peptide is selected for fragmentation[58]. Given the physiological importance of deamidation in CeD pathogenesis, we sought to manually validate all wheat peptides where deamidation was automatically assigned by the PEAKS software, as described below.

The isotopic pattern of the MS1 (precursor ion) spectrum of each peptide assigned as deamidated by PEAKS was inspected in Thermo Xcalibur 3.0. To determine if the assignment was better explained by bona fide deamidation or selection of the second, or in a few cases, third isotopic peak, the theoretical mass spectra of both the native and deamidated peptides were calculated using the MS-Isotope module of ProteinProspector (http://prospector.ucsf.edu/). Then, the isotopic pattern and mass error of the experimentally observed spectra were compared to the theoretical mass spectra for both the native and deamidated forms of the peptide. If the monoisotopic mass of the native peptide was visible along with the non-monoisotopic mass that was selected for fragmentation, then the peptide sequence was manually reassigned from the deamidated to the native sequence. In cases where the monoisotopic mass of the native peptide was not visible, but the experimentally observed mass more closely matched the theoretical non-monoisotopic mass of the native peptide compared to the monoisotopic mass of the deamidated form, then the sequence was also manually reassigned as native. ~90% of the peptide sequences assigned as deamidated by PEAKS were corrected to their native sequences on the basis of these two criteria. In Supplementary Datasets 1–7, peptides whose PEAKS-assigned sequences were manually reassigned as native are clearly denoted in the "Manually Corrected Sequence" column.

For the remaining ~10% of peptides with putative deamidation assignments, the observed masses and isotopic patterns in the MS1 spectra were either equally consistent with the native and deamidated peptide forms, or more consistent with the deamidated form. In these cases, we inspected the MS2 (product ion) spectra for the presence of fragment ions that would allow us to assign the deamidation to a particular Gln residue within the peptide sequence. In all of these cases, we were unable to unambiguously identify a particular Gln residue that had undergone deamidation, as assessed by the presence of multiple site-determining fragment ions. Because the combined MS1- and MS2-level evidence for these peptide sequences were not clearly more consistent with either the deamidated or the native peptide sequences, we did not consider these peptides for downstream data analysis. We cannot exclude the possibility that some peptides were indeed deamidated (or a mixture of native and deamidated forms) but produced insufficient quality spectra to regiospecifically assign the deamidation site. Alternatively, it is possible that the sequences of these peptides were otherwise misassigned by the PEAKS software. For clarity, all excluded peptides are reported in the PEAKS search output in Supplementary Datasets 1–7 and are denoted "N" in the "Included for Data Analysis" column.

We acknowledge that this approach for validating deamidated peptide sequences is stringent. To assign a peptide as deamidated, we required that the isotopic pattern and mass errors of the MS1 spectra to be consistent with the deamidated form, and for the MS2 spectra to be of sufficient quality to allow clear assignment of the deamidation site. Future studies utilizing targeted mass spectrometry and synthetic peptide standards should be insightful in conclusively determining if deamidated wheat peptides are detectable in urine, and if so, how their concentration compares to the corresponding native forms.

**Statistical analyses and other software**. All statistical tests indicated in the figure legends were performed using GraphPad Prism version 9. Parts of some figures in this manuscript were created using BioRender.com.

**Reporting summary**. Further information on research design is available in the Nature Research Reporting Summary linked to this article.

## Data availability

The mass spectrometry data have been deposited to the ProteomeXchange Consortium via the PRIDE partner repository (https://www.ebi.ac.uk/pride/) under accession number PXD031048. The processed peptide identifications are available in Supplementary Datasets 1–7. The processed data used to generate figures are provided in the Source Data file. Source data are provided with the paper. Any remaining data are available within the Article or from the authors upon request. Source data are provided with this paper.

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

## Acknowledgements

We thank Elvi Sanjines and Gotzone Garay for support with patient recruitment, and Dr. Carlos Gonzales, Dr. Niclas Olsson, and Shelley Dutt for assistance with mass spectrometry. We also thank Dr. Emma Chory for helpful discussions on data processing and visualization. C.K. and B.J received support for this research from the National Institutes of Health under award number NIH R01 DK063158. C.K. gratefully acknowledges seed funding from The Joint Institute for Metrology in Biology, founded by the National Institute of Standards and Technology. J.E.E. is supported by the Chan Zuckerberg Biohub. N.F.B. gratefully acknowledges Joelle and Robert Triebsch and the Division of Gastroenterology and Hepatology for supporting our celiac translational research program. C.S. was supported by a grant from Ministerio de Ciencia e Innovación from Spain and FEDER funds (SAF2017-83700-R).

## Author contributions

B.A.P., N.W., L.Z., A.J.H., B.J., N.Q.F., C.K., and J.E.E. designed research. B.A.P., N.W., L.Z., A.J.H., L.A.F., and K.S. performed research. C.S. contributed reagents and specimens. B.A.P., N.W., L.Z., A.J.H., C.K., and J.E.E. analyzed the data. B.A.P., N.W., A.J.H., C.K., and J.E.E. wrote the paper with input from all authors.

## Competing interests

C.K. is a member of the Board of ImmunogenX, a company developing celiac disease therapies. The remaining authors declare no competing interests.
