## [Peer Review File · Nature Communications]

Reviewer comments, initial round review:

Reviewer #1 (Remarks to the Author):

In their manuscript entitled "An efficient urine peptidomics workflow identifies chemically defined dietary gluten peptides from patients with celiac disease" Palanski et al. developed a liquid chromatographic-mass spectrometric workflow for untargeted sequence analysis of urinary peptides. The authors subsequently applied the workflow and report the tentative identification of 679 distinct dietary peptides.

In principle this may be a relevant topic, but I have several major concerns regarding the manuscript:

1. In the introduction the authors argue that "established LC-MS/MS methods suffer from severe technical limitations when applied to urinary peptidome analysis". However, multiple authors have described exactly this, urinary peptidome analysis using either LC-MS/MS or CE-MS/MS. Unfortunately, these papers, in cases reporting on the investigation of thousands of samples, are not mentioned. I suggest to review the literature on the topic.

2. The authors proceeded to develop a workflow that should enable enrichment of gluten peptides. As one of the first steps, reproducibility and fold enrichment should be assessed, and data should be presented along these lines, e.g. results from a sample without and with enrichment applied. I was not able to find such data.

3. Another concern is that most peptides were found only in one sample. This generates concerns that the results may to a large degree be artefacts and/or erroneous interpretation of the MS/MS spectra (see also the issue on collagen and hydroxyproline, below). Substantially more consistency between the different samples would be expected. To support that in fact the results correctly reflect the actual content in gluten peptides, at least reproducibility of the technical approach, based on repeat preparation and analysis of the same sample, should be assessed.

4. Based on the literature I am aware of, a large part, actually the majority of urine peptides are collagen derived, containing as post translational modification hydroxyproline. Based on the methods disclosed the authors did not account for this modification, which in turn would result in the inability to identify these sequences correctly (and likely the assignment of incorrect sequence at least for some, if not for a large number of spectra).

5. The number of subjects included in the study is extremely low. In combination with the fact that most peptides were only found in one sample, indicating very high variability, this generates the impression that the data presented are not representative.

Additional comments:

In general, the manuscript is hard to read, based on all the different data, supplements, tables, and figures.

There is a contradiction of the sentence: "In fact, not a single chemically defined peptide from wheat (or, to our knowledge, from any dietary protein) has ever been identified from the human circulatory or excretory systems." with the statement that "over 20 years ago, chromatographic analysis implied the existence of gluten-derived peptides in CeD patients' urine."

All the different methods used (and ultimately found of no value) for LC-MS/MS analysis likely are of no substantial interest for the readers. Therefore, I suggest excluding all information of the not-successful methods (including tables, methods, and results) and only presenting the methods that appeared to be of value (however, please see comments above).

Why the authors did not use the already existing and established sample preparation method for urine peptides? This method is already used for almost 20 years and resulted in the identification of several thousand peptides.

Statistical analysis is not described, in fact it seems statistics was not applied.

Reviewer #2 (Remarks to the Author):

In their manuscript entitled „An efficient urine peptidomics workflow identifies chemically defined dietary gluten peptides from patients with celiac disease“, Palanski et al describe a novel untargeted LC-MS based workflow to efficiently characterize urinary peptides.

The manuscript is very well written and results are clearly described. The methods are scholarly described in all detail, and all rawdata have been submitted to a public repository. The workflow described by the authors is highly efficient and enabled the author to identify close to 700 diet-derived peptides in the urinary peptidome. By including a dietary negative control, the authors prove that peptides are indeed derived from the diet, in this case gluten and related wheat/barley/rye proteins. Of note, the peptides contain known celiac disease-related epitopes and some of the peptides are even known to elicit innate immune responses. In addition to describing a novel untargeted LC-MS workflow to analyze the urinary peptidome, the paper is a very important step forward towards understanding the pathophysiology of celiac disease, as it is the first study to identify the specific amino acid sequences and post-translational modifications of the peptides resulting from in vivo digestion of gluten or any other dietary proteins.

Reviewer #3 (Remarks to the Author):

Congratulation for this valuable study.

It is the first study to develop a novel LC-MS/MS-based assay that directly detects the metabolic products of food grains (prolamins). It reveals in the urine, the proteome, the precise chemical structures of dietary peptides that may drive Celiac disease (CeD) in humans.

The findings are novel and very much of interest to the CeD research community. The conclusions are original and based on convincing results. It might represent a game-changer in understanding CeD pathophysiology, GFD compliance follow up and if substantiated on more patients, normal and pathological controls, the study might change the current diagnostic criteria.

Comments:

1. Lines 334-338 Multiple processed food additives increase intestinal permeability and microbial transglutaminase can imitate the tTG deamidation/cross linking of gliadin peptides

Please see: doi: 10.1016/j.autrev.2015.01.009

doi: 10.3390/ijms21031127,

2. Lines 350-360 "unable to definitively identify any peptide that underwent regioselective Gln deamidation by

352 transglutaminase 2" It is known that TG2 can also cross-link gluten/gliadin peptides and post-translationally modify them. Could the TG2 cross-linked gliadin peptides be detected in the urine?

3. A future study might contain CeD patients with and without microbial TGase processed food in order to study their differential urine proteome

4. Please discuss the potential of your results on the extra-intestinal manifestations of CeD pathophysiology

Please, comment on the importance to apply your methodology on blood samples of CeD/controls. Gluten/gliadin peptides cross-react with numerous food products antibodies and have sequence homology to multiple human tissues' antigens, including in the human brain

PMID: 33808124 PMCID: PMC8065505 DOI: 10.3390/cells10040756

Please, discuss those published effects on urinary proteome of CeD patients

Reviewer #4 (Remarks to the Author):

In the present study the authors have reported a novel liquid chromatographic-mass spectrometric workflow for untargeted sequence analysis of the urinary peptidome using a specialized protocol to detect a large number of unique (wheat, rye and barley) peptides in the urine of healthy subjects and patients with celiac disease. The number of peptides detected in the urine of Celiac disease patients was more than that detected in non-celiac individuals.

The detection of wheat peptides in the urine opens new opportunities to develop tests which can be

used for assessment of adherence to gluten-free diet.

1. There is a lot of variation in the detection of peptides in both controls and patients with celiac disease. In one subject with celiac disease, peptide extracted in the urine was some 275, while in others a few only. Is it related with extraction method or patient to patient variation.
2. Is this also related with a small sample size included in all the studies (healthy and patients with CeD
3. Could the differences in the detection of peptides also be accounted by the use of banked samples versus relatively fresh samples.
4. It is unclear, how was creatinine normalization done? Creatinine excretion may vary individual to individual depending upon the muscle mass and renal functions.

Reviewer #1 (Remarks to the Author):

In their manuscript entitled "An efficient urine peptidomics workflow identifies chemically defined dietary gluten peptides from patients with celiac disease" Palanski et al. developed a liquid chromatographic-mass spectrometric workflow for untargeted sequence analysis of urinary peptides. The authors subsequently applied the workflow and report the tentative identification of 679 distinct dietary peptides. In principle this may be a relevant topic, but I have several major concerns regarding the manuscript:

We thank the reviewer for their constructive comments, which have led to a substantial improvement in our work.

1. in the introduction the authors argue that "established LC-MS/MS methods suffer from severe technical limitations when applied to urinary peptidome analysis". However, multiple authors have described exactly this, urinary peptidome analysis using either LC-MS/MS or CE-MS/MS. Unfortunately, these papers, in cases reporting on the investigation of thousands of samples, are not mentioned. I suggest to review the literature on the topic.

It was certainly not our intention to discount prior work on MS analysis of the urinary peptidome. However, the focus of our manuscript is LC-MS/MS and not CE-MS or CE-MS/MS, owing in part to the ubiquity of LC-MS systems compared to CE-MS (like many labs and core facilities, we do not have a CE-MS/MS), but more importantly, the dramatic advantage LC-MS/MS has in sampling depth for peptidomics in comparison to CE-MS/MS. We wanted to develop as broadly accessible and sensitive of a urinary peptidomics method that we could, and therefore, focused on LC-MS/MS. Of the 100s of papers we could have cited, we selected those that identified a substantial number of peptides (100s-1000s from a typical urine sample) by LC-MS/MS. Particularly, we emphasized work that focused on method development and/or removing interfering substances such as the urinary pigment urochrome. Following the reviewer's suggestion, we again searched the literature but did not find more appropriate LC-MS/MS papers than those originally cited.

Specifically, the papers originally cited in our introduction (Main Text refs. 32 and 33) described a yellow-brown substance (presumably urochrome) that cannot be removed with common extraction techniques. We verified that typical reversed phase and/or liquid-liquid extraction procedures did not remove urochrome (Supplementary Fig. 3). In LC-MS/MS, urochrome's presence severely suppresses the number of peptide identifications and leads to instrument downtime (ref. 32 and Supplementary Fig. 3). Two recent papers report a comparable number of urinary peptide identifications to what we achieve with our optimized method (at least 1000s of unique peptides from a typical urine sample). However, both papers (Main Text ref. 43, published in 2017, and Supplementary ref. 6, published in 2020) report a requirement for offline strong cation exchange (SCX) fractionation using an HPLC. In Supplementary ref. 6, after initial solid phase extraction of urine samples, offline SCX "was carried out continuously for 2–3 weeks", and samples still required an additional desalting step prior to LC-MS/MS analysis. The mixed cation exchange solid phase extraction technique we developed eliminates the need for offline SCX, allowing urine samples to be prepared in ~4 hours. In summary, our work overcomes longstanding challenges with literature reported methods for LC-MS/MS urinary peptidomics. Specifically, we achieve removal of urinary pigments without the time, sample loss, or specialized equipment needed for offline SCX, while recovering sufficient peptide to enable identification of a similar number of sequences reported in the current LC-MS/MS urinary peptidome literature.

2. The authors proceeded to develop a workflow that should enable enrichment of gluten peptides. As one of the first steps, reproducibility and fold enrichment should be assessed, and data should

be presented along these lines, e.g. results from a sample without and with enrichment applied. I was not able to find such data.

Regarding **enrichment**, we believe the reviewer reached an unintended conclusion about our experimental methods: We employed no *specific* strategy (e.g., affinity resins) to enrich for gluten peptides. The major technological advance we describe relates to our preparation technique which effectively **depletes** urine of interfering small molecules such as urochrome. Without these inhibitory molecules, our LC-MS/MS analyses yield far deeper surveys of the urine peptidome, enhancing our ability to detect gluten peptides. This enhancement enabled us to measure beyond the top ~200-most abundant urinary peptides that had been the status quo from most prior LC/MS-MS literature (with the exception of a few papers that employed specialized, time-consuming offline SCX HPLC, as detailed in our response to Comment 1). We can measure over 2,000 unique peptides from a single specimen with our optimized procedure (Supplementary Fig. 3). The various optimization attempts that this figure examines revealed that wheat-derived peptides are a minor but consistently detectable portion of the overall urine peptidome (2-7%) from wheat-fed subjects.

The enrichment in gluten peptides that we report is with respect to the less sensitive methods that we initially tested, supporting data for which were included in our original draft (presented in Supplementary Figs. 2-3 and summarized in lines 129-133 of the Main Text). In Supplementary Fig. 2, we use a mass spectrometry independent technique (ELISA) to optimize initial recovery steps. We show that urine sample processing with a 10 kDa molecular weight cutoff centrifugal filter improves gluten peptide recovery **15-fold** relative to total urinary proteins and peptides. Next, we use LC-MS/MS to directly identify gluten peptides that are enriched from urinary proteins, metabolites, and salts using the steps optimized in Supplementary Fig. 2 in combination with various downstream processing methods. Relative to the initial method (Method A), the optimized enrichment method (Method E) identifies **>31-fold** more gluten peptide sequences (Supplementary Figure 3). We hypothesize that the dramatically improved performance of Method E results from improved gluten peptide recovery (Method E has the fewest steps, providing less opportunity for sample loss) and its ability to effectively remove urochrome, reducing ion suppression.

We recognize that the statement, “Therefore, we developed an extraction technique to enrich urinary wheat peptides while removing these interfering compounds (Supplementary Fig. 3)” (line 301 in our original draft) may have caused misconceptions and confusion about our method. We hope the above description has clarified that by “enrichment,” we originally meant overall peptide enrichment from urinary salts, proteins, and metabolites such as urochrome. However, we have now broadly edited the text to describe “overall peptide enrichment”, or “gluten peptide recovery” to avoid the impression that our method specifically targeted gluten peptides above all others. The statement above now reads, “Therefore, we developed an extraction technique to remove these interfering compounds (**Supplementary Fig. 3**), and dramatically improve the ability to measure all urinary peptides, including those derived from wheat.” (lines 303-305).

During initial workflow development, our primary goal was to uncover overall trends in gluten peptide recovery and identification. While arriving at Method E, we found logical trends in the factors that improved overall peptide enrichment (Supplementary Figs. 2-3). After identifying Method E as the preferred method, we next chose to test whether it was reliable in differentiating urine from gluten challenged versus gluten fasted individuals (Main Text Figure 2). Furthermore, we tested whether it could accurately identify the sequences of diet-derived peptides (Main Text Figure 3). (See also our response to Comment 3, Points (a) and (b) below). After validating that Method E reliably identified dietary peptides, we tested this method’s **reproducibility** using the

samples most central to our findings (Main Text Figure 5). However, we apologize that although we conducted repeat preparation and analysis (as requested by the reviewer) for these critical samples, we did not emphasize this point outside of the Materials and Methods and Supplementary Datasets. Our data show good technical reproducibility that is in line with expectations for data-dependent LC-MS/MS runs. In agreement with the reviewer's suggestion that these data are important and may be of interest to readers, in the revised manuscript, we provide a more detailed reproducibility analysis. This is also summarized below for the reviewer's convenience.

As originally described in Materials and Methods lines 459-461, the urine samples in our prospective clinical study were prepared in duplicate on separate days and analyzed by LC-MS/MS in independent runs. In the new version, we added to the Figure 5 caption that "In b-h, all samples were analyzed in duplicate, and the aggregated results are shown. Analyses of individual replicates are provided in Supplementary Figure 11".

Supplementary Figure 11 (reproduced here for convenience). Technical reproducibility of peptide identifications. Urine samples from CeD patients (n=6), patients with non-CeD gastrointestinal disorders (n=5) or healthy controls (n=8) were independently prepared on separate days and analyzed in independent LC-MS/MS runs. (a) Human peptide sequences identified in Replicate 1. (b) Human peptide sequences identified in Replicate 2. (c) Human peptide sequences identified only in both replicates. (d) Human peptide sequences identified only in both replicates expressed as a percentage of total peptides. (e) Wheat peptide sequences identified in Replicate 1. (f) Wheat peptide sequences identified in Replicate 2. (g) Wheat peptide sequences identified only in both replicates. (h) Wheat peptide sequences identified only in both replicates expressed as a percentage of total peptides. Samples with fewer than 20 wheat peptides identified in either replicate are denoted as gray circles. *p < 0.05; **p < 0.01. One-way Kruskal-Wallis ANOVA/Dunn's multiple comparison test.

When the replicate data (independent preparations and LC-MS/MS runs of the same urine sample) are analyzed separately, the number of human (Supplementary Fig. 11a,b), and wheat (Supplementary Fig. 11e,f) peptide sequences identified in each individual's urine is similar. Importantly, the differences in the number of identified wheat peptide sequences between CeD patients, patients with non-CeD gastrointestinal disorders (n=5), and healthy controls are statistically significant in either replicate alone (Supplementary Fig. 11e,f), while the number of human peptide sequences do not significantly differ in either replicate (Supplementary Fig. 11a,b). These results strongly support the reproducibility of our main conclusion that the peptide repertoires of CeD patients significantly differ from healthy controls.

In Main Text Fig. 5b, we reported the aggregated results from both replicates in order to maximize peptide sequence identification, as is common in data-dependent MS experiments, where peptide sampling is stochastic. In the revised manuscript, we also applied a more conservative analysis where we only counted peptides that were directly identified by their MS/MS spectra in both replicates. Again, the results remain statistically significant for wheat peptides (Supplementary Fig. 11g) but not for human peptides (Supplementary Fig. 11c), in line with all other analyses.

To further aid in visualizing reproducibility, we expressed the percentages of overlapping human (Supplementary Fig. 11d) and wheat (Supplementary Fig. 11h) peptides found in both replicates. For human peptides, the percentage of sequences identified in both replicates spanned 43-68%, with a median value of 57% (Supplementary Figure 11d). For wheat peptides, the percentage of sequences identified in both replicates spanned 12-50%, with a median value of 36% (Supplementary Fig. 11h). The seven samples with the lowest wheat peptide percent reproducibility (denoted as grey data points in Supplementary Fig. 11h) were from Other GI Patients or Healthy Controls that had very few overall wheat identifications (<20 wheat peptides). As expected from a stochastic sampling method, in these samples that contain a very small proportion of wheat peptides compared to the total number of peptides, there is less of a chance of detecting the same wheat peptide in both replicates. When considering samples with a substantial number of wheat peptide identifications the overlap is higher and similar to the overlap observed for human peptides which comprise the majority of the data. Overall, however, the average overlap in identified peptide sequences in our replicates is well within the expected values for data dependent MS runs, where overlap between peptide lists in technical replicates typically spans between 35-60% (Tabb, D.L., et al. (2010). Repeatability and reproducibility in proteomic identifications by liquid chromatography-tandem mass spectrometry. *J. Proteome Res.* 9, 761–776). We wish to emphasize this expected number of 35-60% peptide overlap is based on literature studies in which the same preparation of the same sample was injected multiple times (instrumental technical replicates). In contrast, we analyzed completely separate preparations of the same sample, adding an additional source of variability. Therefore, the fact that the overlap for our preparative technical replicates is within the accepted range for instrumental technical replicates strongly supports that our method is reproducible.

Additionally, to acknowledge the fact that future to *validate* the utility of specific peptides in discerning CeD status would benefit from targeted LC-MS/MS methods (as opposed to this study, whose primary goal was to *discover* such peptides), we added the following sentence to lines 388-390 of the discussion: "Such validation studies should benefit from development of targeted LC-MS/MS methods that reproducibly sample and quantify the peptides of interest."

3. Another concern is that most peptides were found only in one sample. This generates concerns that the results may to a large degree be artefacts and/or erroneous interpretation of the MS/MS spectra (see also the issue on collagen and hydroxyproline, below). Substantially more consistency between the different samples would be expected. To support that in fact the results

correctly reflect the actual content in gluten peptides, at least reproducibility of the technical approach, based on repeat preparation and analysis of the same sample, should be assessed.

Here, three points are raised: (a) an expectation that there would be more biological (human-to-human) consistency in identified peptide sequences, (b) the possibility that the reported sequences are erroneous, and (c) the need to verify reproducibility. We agree with the reviewer that these are all important considerations, and below we address each point in detail.

- (a) Regarding consistency in the identified peptide sequences, based on literature precedent, we respectfully disagree that high interindividual consistency would be expected. As recently reviewed (Walther, B., et al. (2019). GutSelf: Interindividual Variability in the Processing of Dietary Compounds by the Human Gastrointestinal Tract. *Mol. Nutr. Food Res.* 63, e1900677), it is known that there is high variability in human protein digestion attributable to many factors, from extent of chewing to gastrointestinal protease levels to genetic variation in intestinal peptide transporters. Perhaps most relevant to our work is a study that analyzed protein digestion via MS analysis of small intestinal contents (Boutrou, R., et al. (2013). Sequential release of milk protein-derived bioactive peptides in the jejunum in healthy humans. *Am. J. Clin. Nutr.* 97, 1314–1323). Jejunal contents were directly extracted via nasogastric sampling. Therefore, this analysis revealed peptide processing by only oral and gastrointestinal proteases. In contrast, in our study, peptides were analyzed after additional processing by intestinal brush-border proteases, absorption, and processing by blood, extracellular, and urinary proteases. Despite the substantially simpler level at which peptide digestion was probed in prior published works, high variability was also observed. For example, of the 218 peptides identified from β -casein in 7 individuals sampled at multiple time points, only 11 peptides (~5% of total peptides) were observed with a frequency >60%, and 112 peptides were identified with a frequency of <5%. This is exactly in line with our results that only a small number of peptides make up the “core” urinary diet-derived peptidome, with 24/435 (~5% of total peptides) sequences being detected in the majority of individuals in our clinical study (Main Text Fig. 5 and Supplementary Dataset 7).

In addition to established high variability in protein digestion in general, CeD itself is a highly heterogeneous disease in terms of symptoms and extent of gastrointestinal damage. This heterogeneity likely also contributes to the high degree of interindividual variability observed in CeD patients. We also note that although the exact peptide sequences are variable, many of the wheat peptide sequences found in different individuals differ by only a few amino acids at either terminus. These “ladder peptides” map to the same regions of the wheat proteome (Main Text Fig. 5g-j and Supplementary Fig. 10). If the peptides were artifacts, then we would expect them to map randomly to the wheat proteome. In the revised manuscript, we added additional examples of this peptide-to-protein mapping to Main Text Fig. 5g-j to demonstrate this consistency more clearly.

Moreover, we have recently obtained antibody data (ELISA) against one of the gluten peptides. Our preliminary data with that antibody confirms a wide (>1-log) inter-individual distribution of abundance of the peptide. While this project is ongoing, our findings with an unrelated method reinforce the notion that heterogeneous processing of dietary peptides between individuals is not an artifact of our LC-MS/MS method.

As a reader may have similar questions as the reviewer about the apparently wide variation in the number of detected peptide sequences, we have added the following statement to discussion lines 382-384:

“This broad distribution of detected peptide sequences likely stems from the intrinsically high variability of human protein digestion⁵³, as well as the fact that CeD is a highly heterogeneous disease in terms of symptoms and extent of gastrointestinal damage⁵⁴.”

Parenthetically, we have corrected a typographical error in line 278 of the Main Text, which previously stated that 283 of the 293 peptides specific to patients with CeD were found only in one individual. The actual number of peptides found in only one individual is 206, as was correctly displayed in Supplementary Table 6 of the original manuscript (Supplemental Dataset 8 of the revised version). We apologize if this error contributed to the reviewer’s impression of extreme variability. We have carefully verified that no similar errors exist in the rest of the manuscript.

- (b) Regarding the possibility that our reported sequences could be artifacts or result from erroneous interpretation of the MS/MS spectra, we present several lines of experimental evidence that that our reported sequences are not artifacts and that our method for interpreting MS/MS spectra is reliable.

We controlled false peptide identifications by using a target-decoy fusion approach with a 1% false discovery rate (FDR) cutoff typically used in the proteomics studies. The experimental data in Fig. 2 provide evidence that our FDR control worked as intended: Wheat peptide sequences were detected in all individuals after wheat dietary challenge. When adherent to a gluten-free diet or after overnight fasting, wheat peptides were very scarcely identified from the same individuals. Specifically, in total in Fig. 2, we identified 372 wheat peptides in 8 individuals. Only 4 peptides (1.1%) were identified in non-gluten challenged urine, and these peptides had low identification (-10lgP) scores close to the cutoff we used to achieve an aggregate 1% FDR for the entire data set, as was noted in Supplementary Table 3 of the original manuscript (now Supplementary Datasets 3 and 4). This indicates our FDR control is reliable. If the observed wheat peptides were largely artifacts, then they would have been more equally distributed between the gluten-free and gluten-challenged urines.

Moreover, our data in Main Text Fig. 3 provide strong evidence that the identified sequences are accurate and not results of erroneous MS/MS spectral interpretation. Wheat, barley, and rye are grains with closely homologous peptide sequences (Main Text Fig. 3a). When two individuals were challenged with wheat, barley, or rye diets on separate occasions, not a single grain peptide sequence was found in common between the wheat, barley, and rye diets (Fig. 3b and Supplementary Table 4 of the original manuscript, now Supplementary Dataset 6). For the reviewer’s convenience, a condensed representation of the data in Supplementary Table 4 is provided below (Fig. R1). Consistent with Main Text Fig. 1, there is considerable interindividual variability in the number of detected peptide sequences. However, of the 132 peptides identified, only 1 sequence (0.75%) was found in gluten-free control urine, again suggesting the FDR was well-controlled. Additionally, as discussed in the Main Text, lines 215-224, these sequences mapped to the respective proteomes of the food grain that was consumed (i.e., in the wheat diet, only wheat peptides were identified). Taken together, these data clearly substantiate that our MS/MS data accurately identifies grain peptide peptides, even when closely homologous sequences simultaneously considered. If this were not the case, then the peptide sequence identifications would not have depended exclusively on source of the dietary grain, and they would not have mapped back to the proteome of the diet consumed.

Fig. R1. Analysis of grain peptide sequences found in the urine of two healthy participants (HPs) challenged with wheat, rye, and barley diets. Each row represents a grain peptide sequence, and columns are shaded black if that sequence was detected in a particular participants' urine. All grain peptides were uniquely found in a single diet. The sequences of these peptides are reported in Supplementary Dataset 6.

Last, we wish to emphasize that we have transparently reported all processed data, including identification scores ($-10\lg P$) in Supplemental Datasets S1-7, so readers can directly judge the confidence in identification of particular peptide sequences that may interest them. Moreover, all raw data has been deposited in the publicly available PRIDE database should a reader want to undertake further examination of the spectra.

- (c) To the point of reproducibility, we apologize that although we conducted repeat preparation and analysis (as requested by the reviewer) for the capstone samples in our study, we did not emphasize this point outside of the Materials and Methods and Supplementary Datasets. Our data show good reproducibility that is in line with expectations for data-dependent LC-MS/MS runs. These data also address the reviewer's Comment # 2 above, but for his/her convenience we have pasted our response below.

As originally described in Materials and Methods lines 459-461, the urine samples in our prospective clinical study were prepared in duplicate on separate days and analyzed by LC-MS/MS in independent runs. In the new version, we added to the Figure 5 caption that "In b-h, all samples were analyzed in duplicate, and the aggregated results are shown. Analyses of individual replicates are provided in Supplementary Figure 11".

Supplementary Figure 11 (reproduced here for convenience). Technical reproducibility of peptide identifications. Urine samples from CeD patients (n=6), patients with non-CeD gastrointestinal disorders (n=5) or healthy controls (n=8) were independently prepared on separate days and analyzed in independent LC-MS/MS runs. (a) Human peptide sequences identified in Replicate 1. (b) Human peptide sequences identified in Replicate 2. (c) Human peptide sequences identified only in both replicates. (d) Human peptide sequences identified only in both replicates expressed as a percentage of total peptides. (e) Wheat peptide sequences identified in Replicate 1. (f) Wheat peptide sequences identified in Replicate 2. (g) Wheat peptide sequences identified only in both replicates. (h) Wheat peptide sequences identified only in both replicates expressed as a percentage of total peptides. Samples with fewer than 20 wheat peptides identified in either replicate are denoted as gray circles. *p < 0.05; **p < 0.01. One-way Kruskal-Wallis ANOVA/Dunn's multiple comparison test.

When the replicate data (independent preparations and LC-MS/MS runs of the same urine sample) are analyzed separately, the number of human (Supplementary Fig. 11a,b), and wheat (Supplementary Fig. 11e,f) peptide sequences identified in each individual's urine is similar. Importantly, the differences in the number of identified wheat peptide sequences between CeD patients, patients with non-CeD gastrointestinal disorders (n=5), and healthy controls are statistically significant in either replicate alone (Supplementary Fig. 11e,f), while the number of human peptide sequences do not significantly differ in either replicate (Supplementary Fig. 11a,b). These results strongly support the reproducibility of our main conclusion that the peptide repertoires of CeD patients significantly differ from healthy controls.

In Main Text Fig. 5b, we reported the aggregated results from both replicates in order to maximize peptide sequence identification, as is common in data-dependent MS experiments, where peptide sampling is stochastic. In the revised manuscript, we also applied a more conservative analysis where we only counted peptides that were directly identified by their MS/MS spectra in both replicates. Again, the results remain statistically significant for wheat peptides (Supplementary Fig. 11g) but not for human peptides (Supplementary Fig. 11c), in line with all other analyses.

To further aid in visualizing reproducibility, we expressed the percentages of overlapping human (Supplementary Fig. 11d) and wheat (Supplementary Fig. 11h) peptides found in both replicates. For human peptides, the percentage of sequences identified in both replicates spanned 43-68%, with a median value of 57% (Supplementary Figure 11d). For wheat peptides, the percentage of sequences identified in both replicates spanned 12-50%, with a median value of 36% (Supplementary Fig. 11h). The seven samples with the lowest wheat peptide percent reproducibility (denoted as grey data points in Supplementary Fig. 11h) were from Other GI Patients or Healthy Controls that had very few overall wheat identifications (<20 wheat peptides). As expected from a stochastic sampling method, in these samples that contain a very small proportion of wheat peptides compared to the total number of peptides, there is less of a chance of detecting the same wheat peptide in both replicates. When considering samples with a substantial number of wheat peptide identifications the overlap is higher and similar to the overlap observed for human peptides which comprise the majority of the data. Overall, however, the average overlap in identified peptide sequences in our replicates is well within the expected values for data dependent MS runs, where overlap between peptide lists in technical replicates typically spans between 35-60% (Tabb, D.L., et al. (2010). Repeatability and reproducibility in proteomic identifications by liquid chromatography-tandem mass spectrometry. *J. Proteome Res.* 9, 761–776). We wish to emphasize this expected number of 35-60% peptide overlap is based on literature studies in which the same preparation of the same sample was injected multiple times (instrumental technical replicates). In contrast, we analyzed completely separate preparations of the same sample, adding an additional source of variability. Therefore, the fact that the overlap for our preparative technical replicates is within the accepted range for instrumental technical replicates strongly supports that our method is reproducible.

Additionally, to acknowledge the fact that future to *validate* the utility of specific peptides in discerning CeD status would benefit from targeted LC-MS/MS methods (as opposed to this study, whose primary goal was to *discover* such peptides), we added the following sentence to lines 388-390 of the discussion: “Such validation studies should benefit from development of targeted LC-MS/MS methods that reproducibly sample and quantify the peptides of interest.”

4. Based on the literature I am aware of, a large part, actually the majority of urine peptides are collagen derived, containing as post translational modification hydroxyproline. Based on the methods disclosed the authors did not account for this modification, which in turn would result in the inability to identify these sequences correctly (and likely the assignment of incorrect sequence at least for some, if not for a large number of spectra).

In our initial data analysis, we chose the variable modifications based on post-translational modifications that are known to influence the immunogenicity of wheat peptides. The focus of our manuscript is to identify dietary and not host-derived peptides such as collagen. Nonetheless, we thank the reviewer for raising the possibility that failing to account for an abundant modification could cause peptide misidentification. Because our searches are unrestricted/nonspecific with respect to proteolysis (i.e., cleavage is allowed after any amino acid) and already contained several variable modifications, adding hydroxyproline increased the search space and runtime. Therefore, to assess whether this potentially important issue affected our findings, we reanalyzed a subset of our data central to our key findings. Including hydroxyproline as a variable modification did not impact identification of wheat-derived peptides or any of our other findings in a substantiative manner.

The number of wheat or human-derived peptide sequence identifications in CeD patients or patients with non-celiac GI disorders did not significantly change when hydroxyproline was included in the search (Fig. R2).

Fig. R2. Inclusion of hydroxyproline in database searching does not significantly affect the number of (a) wheat or (b) human peptide identifications. Data were analyzed using PEAKS software with the parameters as described in the Main Text (● black circles) or with identical parameters except the addition of hydroxyproline as a variable modification (■ blue squares). ns, not significant (unpaired t-test).

To further assess potential spectral misassignments, we investigated differences in peptide sequence identifications by concatenating identifications from all individuals (Fig. R3). The reviewer is correct that including hydroxyproline as a variable modification increased the number of collagen peptide identifications. Of the human peptides unique to searches including hydroxyproline, 1530/2396 were collagen derived. Nonetheless, the vast majority of human peptides (15480/18734) were identified regardless of whether hydroxyproline was allowed as a variable modification (Fig. R3a). More important to our main goal of dietary peptide identification, similar results were obtained when wheat peptides were analyzed. 397/532 wheat peptides were identified in both searches (Fig. R4b). Additionally, detailed analysis of the 446 wheat peptides reported in our original dataset revealed that not a single spectrum was assigned to a different sequence when hydroxyproline was included in the search. The small differences in overall peptide identifications result from identification of hydroxyproline-containing peptides and slightly different distributions of identification ($-\log P$) scores, which altered the cutoff for maintaining a 1% FDR.

Fig. R3. Identified peptide sequences are consistent regardless of inclusion of hydroxyproline as a variable modification. Overlap between (a) human and (b) wheat peptide sequences. The sequences were concatenated from all CeD patients and Other GI Patients shown in Fig. R2. Data were analyzed using PEAKS software with the parameters as described in the Main Text or with identical parameters except the addition of hydroxyproline as a variable modification.

In summary, inclusion of hydroxyproline as a variable modification did not significantly alter the number of human or wheat peptide identifications or the identified sequences themselves. Moreover, the difference in wheat peptide identifications between two key groups in our study (CeD patients and non-CeD GI patient controls) remained virtually identical. We hope the reviewer appreciates that undertaking even this limited data reanalysis took several weeks of computational time. Reanalyzing all our data in this way would markedly delay dissemination of our work without impacting the key results. Our raw and processed data are deposited in a public repository (PRIDE). Therefore, if other researchers have interest in particular dietary or host-derived peptides, they can reanalyze our data according to their specific needs. In the revised version of the manuscript, we have emphasized in discussion lines 312-316 that these data are available for reprocessing using other PTMs. The new statement is also pasted below:

“Although here we focused on identification of wheat derived peptides, we also identified over 30,000 human peptides (Supplementary Datasets 1-7), which is to our knowledge, the largest collection of urinary peptides sequenced by LC-MS/MS to date. Moreover, we have deposited the raw data in the PRIDE database to allow additional analyses (e.g., using other variable modifications or alternative search engines).”

5. The number of subjects included in the study is extremely low. In combination with the fact that most peptides were only found in one sample, indicating very high variability, this generates the impression that the data presented are not representative.

Subjects were recruited from patients undergoing evaluation for celiac disease at the Celiac Disease Program at the Stanford Digestive Health Center. The recruitment period spanned two years. The criteria for inclusion were (1) symptoms suggestive of celiac disease (e.g., dyspepsia, bloating and diarrhea) but no prior diagnosis and (2) gluten-containing diet status and willingness to undergo a defined dietary gluten challenge. We hope that the reviewer can understand how these inclusion criteria made it difficult to accrue a larger number of patients. We were limited to the small subset of patients with symptoms consistent with celiac disease who were following a normal, gluten-containing diet. Many patients who met the inclusion criteria chose to avoid gluten regardless of whether the diagnosis was confirmed by the diagnostic standard and were unwilling

to consume two bagels prior to urine collection. Furthermore, urinary collection was not universally feasible for all potential candidates, due to logistical constraints. In addition, we could not recruit patients with a previous CeD diagnosis who had already initiated a gluten-free diet, because these patients often have dramatically varying degrees of mucosal healing, depending on initial disease severity and length of/strictness of adherence to the gluten-free diet. Thus, to avoid these potentially confounding factors, we limited our recruitment to active, newly diagnosed CeD patients.

Despite extensive efforts by our clinical research coordinators, ultimately, we were only able to gather the urine samples shown in Main Text Fig. 5. Each and every clinical sample that was collected over the two-year recruitment period was analyzed and included in the data analysis. No samples were excluded for any reason. Furthermore, the clinical research coordinators blinded the investigators from the participants' celiac disease status until after the urine had been analyzed and the data had been processed. Despite the relatively small sample size, our data reached statistical significance. As noted by reviewers 2, 3, and 4, these data are extremely interesting to the CeD research field. We hope that this manuscript will motivate other celiac disease centers to contribute samples to our research in the future, and/or adopt this method independently. We agree with the reviewer that larger sample cohorts will be needed to fully understand the wide range of peptides found in this study and have noted this point in the discussion lines 386-387:

“Undoubtedly, future studies using larger sample cohorts will be required to identify and validate strong candidate peptides for diagnostic purposes.”

Additional comments:

In general, the manuscript is hard to read, based on all the different data, supplements, tables, and figures.

We apologize that the reviewer found the organization of the manuscript confusing. We have reorganized the manuscript and supplement according to the Nature Communications guidelines provided by the editor and hope that this has improved the readability.

There is a contradiction of the sentence: “In fact, not a single chemically defined peptide from wheat (or, to our knowledge, from any dietary protein) has ever been identified from the human circulatory or excretory systems.” with the statement that “over 20 years ago, chromatographic analysis implied the existence of gluten-derived peptides in CeD patients' urine.”

We thank the reviewer for pointing out that we did not explain this point well enough in the original version of the manuscript. Although chromatographic analysis implied that gluten peptides were present in CeD patients' urine, this work was conducted before mass spectrometry was widely interfaced to liquid chromatography. The authors of Main Text ref. 25 showed that celiac disease patients had increased quantities of urinary peptides, as judged by UV detection of chromatographic peaks at 215nm and 280nm indicating peptide bonds and aromatic amino acids, respectively. However, this technique could not identify the amino acid sequences and post-translational modifications of these peptides. Thus, the peptides were not chemically defined. We have rephrased the background leading to this statement in lines 74-82 to make this clearer. Additionally, we now explicitly associate ref. 25 with the first sentence of this paragraph to help the reader identify relevant literature. In the previous version we inadvertently grouped Main Text ref. 25 with the next group of citations on antibody-based methods. The revised text is pasted below:

“Over 20 years ago, chromatographic analysis **coupled to UV detection** implied the existence of gluten-derived peptides in the urine of patients with CeD²⁵. This was confirmed more recently by antibody-based method^{26–29}. Indeed, most current gluten detection methods rely on monoclonal antibodies, which recognize amino acid motifs present in a subset of gluten proteins³⁰. Notwithstanding the valuable knowledge that has been gained from analyzing biospecimens with these immunoreagents, they are neither **capable of revealing the exact gluten peptide sequences that are present**, nor are they comprehensive in that some CeD-relevant peptides may lack the motifs these antibodies recognize.”

All the different methods used (and ultimately found of no value) for LC-MS/MS analysis likely are of no substantial interest for the readers. Therefore, I suggest excluding all information of the not-successful methods (including tables, methods, and results) and only presenting the methods that appeared to be of value (however, please see comments above). Why the authors did not use the already existing and established sample preparation method for urine peptides? This method is already used for almost 20 years and resulted in the identification of several thousand peptides.

We respectfully disagree with the reviewer’s assessment that this portion of our manuscript would be of little interest. Before initiating this work, we surveyed the literature in great depth. We found well-established methods for LC-MS/MS analysis of the urinary *proteome* (i.e., large polypeptides >10kDa) but few LC-MS/MS methods for analyzing the urinary *peptidome* (i.e., relatively small, naturally occurring polypeptides <5kDa). As described in the Main Text and elaborated in the supplement, we did try to apply already existing urinary peptidomic methods. As exemplified in Supplementary Fig. 1, we found that our overarching goal of discovering dietary gluten peptides from urine could not be achieved. Specifically, in agreement with Main Text refs. 32 and 33, methods relying on traditional solid phase and/or liquid-liquid extractions were unable to separate urinary pigments from peptides. These urinary metabolites led to instrument downtime and severely suppressed peptide identifications. Discussions with our colleagues have suggested there is a great deal of interest in urinary peptidomic analysis for other applications, such as understanding food allergy and kidney function. Here, the major goal of method optimization was to achieve maximal gluten peptide detection. For other applications, further refinement may be desirable. In the spirit of open science and transparency, we feel that including full details would be most helpful to other groups seeking to adapt our method.

Statistical analysis is not described, in fact it seems statistics was not applied.

Our statistical analysis is described in the figure captions. In this revised manuscript, we have carefully verified that all statistical analyses are reported following the Nature Portfolio Reporting Summary policies.

Reviewer #2 (Remarks to the Author):

In their manuscript entitled „An efficient urine peptidomics workflow identifies chemically defined dietary gluten peptides from patients with celiac disease”, Palanski et al describe a novel untargeted LC-MS based workflow to efficiently characterize urinary peptides.

The manuscript is very well written and results are clearly described. The methods are scholarly described in all detail, and all rawdata have been submitted to a public repository. The workflow described by the authors is highly efficient and enabled the author to identify close to 700 diet-derived peptides in the urinary peptidome. By including a dietary negative control, the authors prove that peptides are indeed derived from the diet, in this case gluten and related

wheat/barley/rye proteins. Of note, the peptides contain known celiac disease-related epitopes and some of the peptides are even known to elicit innate immune responses. In addition to describing a novel untargeted LC-MS workflow to analyze the urinary peptidome, the paper is a very important step forward towards understanding the pathophysiology of celiac disease, as it is the first study to identify the specific amino acid sequences and post-translational modifications of the peptides resulting from in vivo digestion of gluten or any other dietary proteins.

We thank this reviewer for his or her positive assessment of our work.

Reviewer #3 (Remarks to the Author):

Congratulations for this valuable study. It is the first study to develop a novel LC-MS/MS-based assay that directly detects the metabolic products of food grains (prolamins). It reveals in the urine, the proteome, the precise chemical structures of dietary peptides that may drive Celiac disease (CeD) in humans.

The findings are novel and very much of interest to the CeD research community. The conclusions are original and based on convincing results. It might represent a game-changer in understanding CeD pathophysiology, GFD compliance follow up and if substantiated on more patients, normal and pathological controls, the study might change the current diagnostic criteria.

We thank this reviewer for his or her enthusiastic evaluation of our manuscript. It is also our hope that this work lays a foundation upon which the CeD research community will build fundamentally new understandings of CeD pathogenesis and novel diagnostics. As the reviewer points out and as indicated in our discussion, follow up studies with more patients and controls will be invaluable in this regard.

Comments:

1. Lines 334-338 Multiple processed food additives increase intestinal permeability and microbial transglutaminase can imitate the tTG deamidation/cross linking of gliadin peptides
Please see: doi: 10.1016/j.autrev.2015.01.009, doi: 10.3390/ijms21031127

In agreement with the reviewer's suggestion, we have added a brief discussion of how our method may help to provide support for the ideas in these references. Please see lines 336-340 of the discussion, also pasted below for convenience:

"More fundamentally, analysis of urine from individuals consuming diets with various food additives, such as microbial transglutaminase, should help to address the emerging hypothesis that these additives contain increased levels of immunogenic peptides and/or directly contribute to increases in intestinal permeability^{47,48}.

2. Lines 350-360 "unable to definitively identify any peptide that underwent regioselective Gln deamidation by transglutaminase 2" It is known that TG2 can also cross-link gluten/gliadin peptides and post-translationally modify them. Could the TG2 cross-linked gliadin peptides be detected in the urines?

We thank the reviewer for this insightful question. In principle, mass spectrometry can detect crosslinked peptides. However, we did not find clear evidence for TG2 crosslinked gluten peptides. Generally, identification of crosslinked peptides in untargeted mass spectrometry data is difficult with existing computational tools. The complexities of identifying naturally formed

crosslinked peptides is compounded by the fact that our search space is already broader than in typical proteomics experiments where peptides have defined cleavage sites (e.g., from trypsin digestion). Nonetheless, we agree with the reviewer that identification of such crosslinked peptide complexes would be valuable. We have recently initiated a project to develop an algorithm that can identify such peptides if they exist in urine. We look forward to sharing the results in future publications.

3. A future study might contain CeD patients with and without microbial TGase processed food in order to study their differential urine proteome

We are currently pursuing several lines of follow up studies and look forward to incorporating this interesting suggestion from the reviewer.

4. Please discuss the potential of your results on the extra-intestinal manifestations of CeD pathophysiology.

Presumably, any peptides that ultimately reach the urine must first reach the systemic circulation, and thus they have potential to access organs such as the brain and skin where CeD symptoms are known to manifest in certain individuals. Because we do not have specific data on the distribution of particular peptides into these organs, we are hesitant to make hypotheses about their roles beyond what we have already indicated in the discussion (lines 286-290). However, knowledge of the circulating peptide repertoire should facilitate future studies on specific peptides that may cause immune responses in organs other than the intestine.

5. Please, comment on the importance to apply your methodology on blood samples of CeD/controls

We believe that analysis of blood (serum/plasma) samples from CeD patients and controls will yield important insights complementing our work with the urinary peptidome. Analysis of blood should help to elucidate the pharmacokinetics of gluten absorption, as blood can be sampled at well-defined times after gluten challenge. Additionally, comparison of blood and urine samples should be insightful in understanding how the kidney processes and excretes gluten peptides.

In pilot experiments, we tried to apply our methodology to serum samples. While we can detect peptides, it appears that our method needs substantial optimization to achieve good peptide recovery while removing serum interferents. The challenge presented by serum is removal of abundant proteins, whereas for urine samples our method is optimized to remove small molecule metabolites. We are actively adapting our method to enrich gluten peptides from serum, but it will require some time to finish this effort and acquire blood samples from CeD patients.

6. Gluten/gliadin peptides cross-react with numerous food products antibodies and has sequence homology to multiple human tissues' antigens, including in the human brain
PMID: 33808124 PMCID: PMC8065505 DOI: 10.3390/cells10040756
Please, discuss those published effects on urinary proteome of CeD patients

At the present time, we are unsure how antibody cross reactivity with gliadin/gluten peptides, other food products, and human self-antigens may ultimately affect the urinary peptidome of CeD patients. However, the results of this manuscript directly reveal, for the first time, the gluten peptide sequences that are present in the human body. It is our hope that knowledge of these peptide sequences will provide a focused starting point for investigators looking for cross-reactive gluten peptide sequences that affect aspects of human health, such as neurodegeneration.

Reviewer #4 (Remarks to the Author):

In the present study the authors have reported a novel liquid chromatographic-mass spectrometric workflow for untargeted sequence analysis of the urinary peptidome using a specialized protocol to detect a large number of unique (wheat, rye and barley) peptides in the urine of healthy subjects and patients with celiac disease. The number of peptides detected in the urine of Celiac diseases was more than that detected in non-celiac individuals. The detection of wheat peptides in the urine opens new opportunities to develop tests which can be used for assessment of adherence to gluten-free diet.

We thank the reviewer for their positive, constructive comments on our article. We agree that a major immediate application of this work will be development of improved tests that target abundant urinary gluten peptides for assessment of gluten-free diet adherence. We hope that our answers below text fully address the reviewer's questions.

1. There is a lot of variation in the detection of peptides in both controls and patients with celiac disease. In one subject with celiac disease, peptide extracted in the urine was some 275, while in others a few only. Is it related with extraction method or patient to patient variation.

Analysis of technical duplicates strongly suggests that this is patient-to-patient (biological) variation and not an artifact of the sample extraction or LC-MS/MS analysis. We apologize that we did not make this point clear in the original version of the manuscript. In the revised manuscript, we have added an analysis of the technical duplicates (Supplementary Fig. 11), also pasted and discussed below for the reviewer's convenience.

When the replicate data (independent preparations and LC-MS/MS runs of the same urine sample) are analyzed separately, the number of human (Supplementary Fig. 11a,b), and wheat (Supplementary Fig. 11e,f) peptide sequences identified in each individual's urine is similar. Importantly, the differences in the number of identified wheat peptide sequences between CeD patients, patients with non-CeD gastrointestinal disorders (n=5), and healthy controls are statistically significant in either replicate alone (Supplementary Fig. 11e,f), while the number of human peptide sequences do not significantly differ in either replicate (Supplementary Fig. 11a,b). These results strongly support the reproducibility of our main conclusion that the peptide repertoires of CeD patients significantly differ from healthy controls.

In Main Text Fig. 5b, we reported the aggregated results from both replicates in order to maximize peptide sequence identification, as is common in data-dependent MS experiments, where peptide sampling is stochastic. In the revised manuscript, we also applied a more conservative analysis where we only counted peptides that were directly identified by their MS/MS spectra in both replicates. Again, the results remain statistically significant for wheat peptides (Supplementary Fig. 11g) but not for human peptides (Supplementary Fig. 11c), in line with all other analyses.

Supplementary Figure 11 (reproduced here for convenience). Technical reproducibility of peptide identifications. Urine samples from CeD patients (n=6), patients with non-CeD gastrointestinal disorders (n=5) or healthy controls (n=8) were independently prepared on separate days and analyzed in independent LC-MS/MS runs. (a) Human peptide sequences identified in Replicate 1. (b) Human peptide sequences identified in Replicate 2. (c) Human peptide sequences identified only in both replicates. (d) Human peptide sequences identified only in both replicates expressed as a percentage of total peptides. (e) Wheat peptide sequences identified in Replicate 1. (f) Wheat peptide sequences identified in Replicate 2. (g) Wheat peptide sequences identified only in both replicates. (h) Wheat peptide sequences identified only in both replicates expressed as a percentage of total peptides. Samples with fewer than 20 wheat peptides identified in either replicate are denoted as gray circles. *p < 0.05; **p < 0.01. One-way Kruskal-Wallis ANOVA/Dunn's multiple comparison test.

2. Is this also related with a small sample size included in all the studies (healthy and patients with CeD

Unfortunately, despite accruing samples for approximately two years, ultimately, we were only able to obtain the samples that were analyzed. However, as explained in our response to Comment 1 above, the differences between CeD patients and control groups were statistically significant in both technical replicates and the aggregate dataset, despite this small sample size. The criteria for inclusion were (1) symptoms suggestive of celiac disease (e.g., dyspepsia, bloating and diarrhea) but no prior diagnosis and (2) gluten-containing diet status and willingness to undergo a defined dietary gluten challenge. We hope that the reviewer can understand how these inclusion criteria made it difficult to accrue a larger number of patients. We were limited to the small subset of patients with symptoms consistent with celiac disease who were following a normal, gluten-containing diet. Many patients who met the inclusion criteria chose to avoid gluten regardless of whether the diagnosis was confirmed by the diagnostic standard and were unwilling to consume two bagels prior to urine collection. Furthermore, urinary collection was not universally

feasible for all potential candidates, due to logistical constraints. Despite extensive efforts by our clinical research coordinators, ultimately, we were only able to gather the urine samples shown in Main Text Fig. 5. Every clinical sample that was collected over the two-year recruitment period was analyzed and included in the data analysis. We agree with the reviewer that larger sample cohorts will be needed to fully understand the wide range of peptides found in this study and have noted this point in the discussion lines 386-389:

“Undoubtedly, future studies using larger sample cohorts will be required to identify and validate strong candidate peptides for diagnostic purposes.”

3. Could the differences in the detection of peptides also be accounted by the use of banked samples versus relatively fresh samples.

No, this difference cannot be accounted for by the use of banked versus relatively fresh urine samples. The CeD patient urine samples analyzed in Fig. 4 were indeed banked from a prior study (Main Text ref. 27) and were not collected using our defined gluten challenge protocol (Main Text Figure 2d). It appeared interesting that the CeD patient samples in Main Text Fig. 4 contained more gluten peptides than those from healthy controls in Figs. 2-3, but like the reviewer, we also wanted to determine if variables in these banked samples (other than CeD status) could have affected gluten peptide detection. This is why in Main Text Fig. 5, samples from all groups were collected using the same gluten challenge protocol and frozen on the day of collection. At the end of the study, all samples were defrosted, processed, and analyzed at the same time. Thus, since all samples in Fig. 5 were collected throughout the two year study recruitment period, the differences we see can be attributed to CeD status.

4. It is unclear, how was creatinine normalization done? Creatinine excretion may vary individual to individual depending upon the muscle mass and renal functions.

All urine samples were normalized such that the volume of urine processed contained 30 μmol creatinine. We have added a more detailed description of our procedure for creatinine measurement in lines 462-474 of the revised manuscript (pasted below for convenience):

“Briefly, duplicate aliquots of urine samples were diluted 1:10 in MilliQ water and 15 μL of the diluted samples, or 15 μL of the kit-provided creatinine standard (0-20 mg/dL final concentration, also diluted in water) were added to a 96-well plate followed by 150 μL of alkaline picrate solution. After incubation for 10 min at room temperature, the initial absorbances at 500 nm was determined on a plate reader. The reaction was quenched with 5 μL of the kit-provided acid solution, incubated for 20 minutes, and the final absorbances at 500 nm was measured. Final absorbance values were subtracted from the initial values, and a calibration curve using the creatinine standards was constructed. The creatinine concentrations from the urine samples were calculated based on this curve. If a urine sample reading fell out of the linear range, the measurement was repeated using an appropriate dilution. A volume of urine containing ca. 30 μmol creatinine (1-10 mL for most donors) was neutralized by addition of aqueous 1 M ammonium bicarbonate solution to a final concentration of 50 mM.”

We agree with the reviewer that creatinine excretion could vary from individual to individual based on several factors including as muscle mass and renal function. Many studies have investigated the validity of different methods for normalizing urine volumes, such as measurement of creatinine, specific gravity, and cystatin C. A few examples are cited below:

Adedeji, A.O., *et. al.* (2019). Investigating the Value of Urine Volume, Creatinine, and Cystatin C for Urinary Biomarkers Normalization for Drug Development Studies. *Int. J. Toxicol.* 38, 12–22.

Miller, R.C., *et. al.* (2004). Comparison of specific gravity and creatinine for normalizing urinary reproductive hormone concentrations. *Clin. Chem.* 50, 924–932.

Each of these investigations has revealed advantages and disadvantages for different types of urine concentration normalization. No method appears universally best. Other recent publications on urinary peptidome analysis (e.g., Main Text refs. 31, 32, and 37) have used creatinine to normalize sample loading. Thus, we chose creatinine to be consistent with studies like ours.

Reviewer comments, second round review:

Reviewer #1 (Remarks to the Author):

Regarding comment 1: I do not agree with the claim "our approach overcomes long standing shortcomings...", as these may be shortcomings in some, but for sure not in all applied protocols, as outlined in my previous comments. As such, this claim is not correct. The authors could calm that their approach overcomes shortcomings in the protocol they applied initially.

comment 2: the authors have properly addressed the comment. However, reproducibility is apparently quite moderate. Although no specific data appear presented, it seems that CV is substantially above 20%.

comment 3: the comment was extensively addressed. However, I am still worried about validity of identification. It may be advisable not to report peptides that were identified in only one sample. When comparing the urinary peptide sequences reported in the supplement, it appears there is very poor to no correlation to the findings reported by other groups (compare *Diagnostics* 2020, 10, 1039; or *Proteomics Clin Appl.* 2010 4, 464) . This is very worrying, suggests that the protocol applied in fact does not enable recovery of all peptides, but rather of only a selected fraction. It seems that the data are not accessible in PRIDE, at least there is no entry under the identifier given in the paper. I am certain this is just a mistake, but it does not increase confidence in the validity of the reported findings.

comment 4: the comment was addressed, but ultimately without any consequence in the manuscript. The sequences reported in the supplement do not appear to contain any collagen derived peptides with proline hydroxylated. As also indicated above, according to several other publications these are generally among the most abundant peptides in urine.

comment 5: I can follow the arguments presented by the authors. However, at the same time this indicates that the results are highly preliminary, and especially many of the specific results, the peptide sequences reported, may be individual, not expected to be reproduced.

The other comments were all properly addressed.

Reviewer #2 (Remarks to the Author):

The authors have significantly improved their study in response to the various reviewers' comments.

I highly recommend publication of this excellent manuscript.

Reviewer #4 (Remarks to the Author):

Thank you very much for your kind consideration of the comments of the reviewers and responding to them.

Reviewer #1 (Remarks to the Author):

Regarding comment 1: I do not agree with the claim "our approach overcomes long standing shortcomings...", as these may be shortcomings in some, but for sure not in all applied protocols, as outlined in my previous comments. As such, this claim is not correct. The authors could calm that their approach overcomes shortcomings in the protocol they applied initially.

In recognition of the reviewer's expertise in urinary peptidomics and their opinion that particular protocols have their own sets of strengths and weaknesses, we have revised the wording of the last paragraph of our Introduction to more specifically reflect why limitations of established urinary peptidomic methods precluded us from achieving our key goal of identifying wheat-derived peptides. We hope the editors will find this revision satisfactory. The revised paragraph is pasted below with additions bolded:

As a step toward filling this knowledge gap, we sought to analyze human urine by liquid chromatography coupled to tandem mass spectrometry (LC-MS/MS). Currently, LC-MS/MS is the most widely used technique for peptide sequencing in complex biological samples. However, **many** established LC-MS/MS methods suffer from technical limitations when applied to urinary peptidome analysis. High concentrations of urinary salts and metabolites, **which are not efficiently removed by standard reversed-phase or liquid-liquid extraction procedures**, can overwhelm chromatography systems and interfere with peptide detection^{32,33}. Here, we develop a novel sample preparation and LC-MS/MS method that **utilizes mixed cation exchange solid phase extraction** to exclude these interfering molecules **in a single step**. **This workflow overcomes problems we initially encountered with adapting established methods for urinary peptidomics, such as the need for time-consuming strong cation exchange purification and/or limited depth of peptide sampling**. With it, we can now efficiently identify dietary gluten peptides and report the precise sequences of such peptides in the urine of human volunteers. We also undertake an exploratory clinical study, which revealed wheat-derived peptides that are substantially different in their chemical and biological properties and are differentially found in patients with CeD versus healthy controls. These peptides are attractive candidates for improving CeD diagnosis and for monitoring patient compliance to gluten-free diets. They also set the stage for elucidating mechanisms underlying the anomalous ADME characteristics of gluten and other dietary proteins. More generally, the successful application of our urinary peptidomic workflow to CeD suggests it should be broadly applicable for direct measurement of any endogenous or exogenous peptide present in urine.

Comment 2: the authors have properly addressed the comment. However, reproducibility is apparently quite moderate. Although no specific data appear presented, it seems that CV is substantially above 20%.

We are pleased the reviewer found the comment adequately addressed. Our specific data regarding technical reproducibility are reported in Supplementary Figure 11. The reproducibility of our replicates aligns with expectations for untargeted data-dependent LC-MS/MS (Tabb, D.L., et al. (2010). Repeatability and reproducibility in proteomic identifications by liquid chromatography-tandem mass spectrometry. *J. Proteome Res.* 9, 761–776).

Comment 3: the comment was extensively addressed. However, I am still worried about validity of identification. It may be advisable not to report peptides that were identified in only one sample. When comparing the urinary peptide sequences reported in the supplement, it appears there is very poor to no correlation to the findings reported by other groups (compare Diagnostics 2020,

10, 1039; or Proteomics Clin Appl. 2010 4, 464). This is very worrying, suggests that the protocol applied in fact does not enable recovery of all peptides, but rather of only a selected fraction.

We agree with the reviewer that validating our peptide identifications is essential to our claims. In our extensive initial response and in Figs. 2-3 of the Main Text, we presented multiple lines of experimental evidence that our method accurately identifies even closely related peptide sequences and that our false discovery rate is controlled at 1%. As pointed out by the other reviewers, knowledge of the wheat peptide sequences formed from digestion will enable follow up studies related to various areas of celiac disease biology and immunology. Future studies will best be facilitated by the most comprehensive inventory of wheat peptides available, and therefore we believe that our readership will be best served by reporting all peptides with identification scores that meet a 1% FDR cutoff, as we currently have in the manuscript. We note that in our discussion, we have been careful to avoid overstating any claims about the diagnostic utility or pathophysiological relevance of any specific peptides, and we have emphasized the need for further validation and follow up studies (e.g., lines 366-369, 381-395).

As far as comparison of our work to the urinary peptidomics literature cited by the reviewer, we first wish to point out that any protocol for peptide isolation and cleanup may only cover a selected subset of peptides. Similar to the reviewer's observation that our method may not recover all peptides, we could argue that the cited published papers also only recovered limited peptide subsets. Moreover, we do not claim that our method enables recovery of all peptides; instead, we have fairly stated that our method allows deeper access to the urinary peptidome than prior published protocols and ultimately allowed us to detect wheat-derived peptides. Nonetheless, as detailed in subsequent paragraphs, we carefully considered the cited literature.

In Proteomics Clin Appl. 2010 4, 464, the urinary peptidomes from samples pooled from multiple individuals were characterized using many replicates of the same sample on multiple CE-MS and LC-MS/MS platforms. Both sample collection and preparation differed significantly from our study. Mid-catch "spot" urine samples were collected instead of the pooled 8-hour samples we used. The peptidomes were isolated by using centrifugal filters with a molecular weight cutoff of 20 kDa instead of 10 kDa (which we found optimal; Supplementary Figure 2) and desalting on a size exclusion column instead of solid phase extraction. Only 292 peptides were identified by LC/MS-MS (in contrast to the >30,000 peptides in our study), which is likely at least partially because the published study used an older LTQ-Orbitrap, which has lower sensitivity, slower MS/MS acquisition rate, and low-resolution MS/MS mass accuracy compared to our Orbitrap Fusion Lumos. Additionally, the study used arbitrary cutoffs for conducting replicate analysis with no clear statistical rationale stated for how these cutoffs were chosen. Given the major differences in sample collection and preparation, LC-MS/MS technology, and statistical validation, we do not feel our results can fairly be compared to this work.

While the results in Diagnostics 2020, 10, 1039 were obtained with more similar, modern LC-MS/MS technology compared to our manuscript, there are also major differences in collection, storage, enrichment, and analytical methods. Therefore, differences in peptide identification are expected. For following reasons, a direct comparison may yield limited information:

1. Regarding sample collection, 110 of 127 samples analyzed in the published study were from pregnant women with preeclampsia, a condition noted by the authors to be associated with elevated urinary protein (which may affect the urinary peptidome, perhaps by biasing detection toward peptides derived from the subset of elevated proteins).

2. Regarding sample preparation, in the published study, peptide enrichment and desalting was achieved using size exclusion desalting columns instead of solid phase extraction. Alkylation and reduction were not performed, and not a single cysteine containing peptide was ultimately identified in published work.
3. Regarding data analysis, in our paper, we searched completely nonspecific digests of the human proteome from UniProt, while the Diagnostics paper states that “a small data base was created for identification and semiquantitative analysis of the massive HPLC-MS/MS data”. The database was not provided, and it was stated that “The detailed description of the urinary proteome data base development is out of the scope of this manuscript”. From the brief methods description, this database was constructed by concatenating protein sequences implicated from proteome-level tryptic digests with peptides previously reported to be present in preeclampsia. It is possible that some of the reported peptides in the cited paper are misidentified because sequences were absent from the search database. Thus, differences in the protein sequence databases used for interpreting mass spectra likely contributed to differences in peptide identifications with our study as well.

Unfortunately, the raw data are not available for us to directly compare our data to the cited work, but we did download and analyze the processed peptide identifications to gain some insight into the differences in peptides identified by our study. The key takeaways are:

1. In the published study, an average of 369 peptides were identified from each urine sample. In our study, we identified ~2000-5000 peptides per urine sample. This supports our claim that our method allows deeper access to the urinary peptidome than established methods.
2. Of the 3869 unique peptides identified in the published study, 1524 (~40%) were not shared between multiple samples. Additionally, 3612/3869 (~96%) peptides were found in less than a third of the samples. This is consistent with our findings that the urinary peptidome has an intrinsic high degree of interindividual variability (discussed in extensively in the previous round of review).
3. To compare peptides identified with our method with the published study as fairly as we could imagine, we concatenated search results from the urines of patients with non-CeD GI (n=5) disorders and patients with CeD (n=7). Our searches were performed by allowing similar variable modifications (including proline hydroxylation) to the modifications in the supplement of Diagnostics 2020, 10, 1039. We found we still detected only ~17% of the sequences reported in the published paper. However, the authors reported that they ultimately considered only a small fraction (<10%) of the detected sequences that were “substantially represented” in their diagnostic groups. When this group of substantially represented peptide sequences was extracted from the supplemental information and compared to our data, ~44% of the published sequences, including PTMs were in common with our data set. When only the base peptide sequences (without PTMs) were considered, ~68% of the peptides that were reproducibly detected in the published study were in common with our data set. Therefore, despite the large differences between the design of our study and the published study, we found a substantial degree of overlap at least in the peptides that the authors considered to be robustly detected.

It seems that the data are not accessible in PRIDE, at least there is no entry under the identifier given in the paper. I am certain this is just a mistake, but it does not increase confidence in the validity of the reported findings.

We apologize that the reviewer was unable to access our study in PRIDE. The study was deposited; however, access had not been made public and the data were only accessible by logging in with reviewer's credentials (**username:** reviewer_pxd023160@ebi.ac.uk; **password:** qF2bXyfl). We will make the dataset public upon acceptance of the manuscript.

Comment 4: the comment was addressed, but ultimately without any consequence in the manuscript. The sequences reported in the supplement do not appear to contain any collagen derived peptides with proline hydroxylated. As also indicated above, according to several other publications these are generally among the most abundant peptides in urine.

We thank the reviewer again for this important comment regarding hydroxyproline. As detailed in our initial response, this prompted us to critically reevaluate our data. However, as the reviewer points out, the results were inconsequential to our findings. Although inclusion of hydroxyproline as a variable modification indeed allowed us to identify more collagen-derived peptides, other peptides were not misidentified when hydroxyproline was not included in the analysis. Moreover, while hydroxyproline containing peptides were readily detectable in our data, the number of wheat or human-derived peptides did not substantially change when hydroxyproline was included. We speculate that this is the case because our method samples peptides with greater depth than established literature methods.

In summary, we clearly demonstrated that including hydroxyproline as a variable modification did not impact identification of wheat-derived peptides, false discovery rate control, or any of our other findings in a substantive manner. We again wish to emphasize that because our searches are unrestricted/nonspecific with respect to proteolysis (i.e., cleavage is allowed after any amino acid) and already contained several variable modifications known to occur on wheat peptides (whose identification was our major goal), adding hydroxyproline dramatically increased the search time. Undertaking the analysis of the subset of samples presented in the initial reviewer response required several weeks of computational time. Moreover, the raw data are deposited in PRIDE and we have noted in lines 317-322 of the discussion that the data are available for researchers to reprocess considering any PTM of interest:

“Although here we focused on identification of wheat derived peptides, we also identified over 30,000 human peptides (Supplementary Datasets 1-7), which is to our knowledge, the largest collection of urinary peptides sequenced by LC-MS/MS to date. Moreover, we have deposited the raw data in the PRIDE database to allow additional analyses (e.g., using other variable modifications or alternative search engines).”

With these considerations and, most importantly, our analysis showing inclusion of hydroxyproline is inconsequential to our findings, we strongly believe there is no benefit to conducting further analysis of hydroxyproline containing peptides, as the very long search times they require would markedly delay publication and dissemination of our work, which as indicated by other reviewers, could be potentially “game changing” for the celiac disease research field.

Comment 5: I can follow the arguments presented by the authors. However, at the same time this indicates that the results are highly preliminary, and especially many of the specific results, the peptide sequences reported, may be individual, not expected to be reproduced.

We appreciate that the reviewer follows the arguments we presented in the previous round of review (specifically, that interindividual variability in dietary peptide sequences are biologically expected and preceded in the literature, that celiac disease itself is a heterogeneous disease, and that we recruited as many participants as possible for our study over a two-year period).

Our data do indeed show that many of the peptide sequences are restricted to particular individuals. In the manuscript and the previous response, we show that our method accurately identifies closely related wheat peptide sequences, and that our false discovery rate is well-controlled at 1%. Therefore, in contrast to the reviewer's negative view of this variability, we argue that these results accurately reflect important information related to inter-individual heterogeneity. Furthermore, they illustrate the power of studying wheat protein digestion through urinary peptidomics: we can reveal exactly which peptide sequences are present in a given individual. This is especially important to further our understanding of celiac immunology, given the surprising and recent finding that approximately 50% of T-cell clones from celiac disease patients react to unknown sequences (Main Text ref. 31), as discussed in lines 402-406.

Additionally, our results clearly reveal that there is indeed a subset of peptide sequences that are present in all or almost all individuals (e.g, GQQQPFPPQQPYPQPQPFPS and derivatives). The presence of these common peptide sequences was not anticipated, and current immunoassays to detect the present gluten in urine have worked under the assumption that other peptides with known T-cell epitopes were most logical to target with antibody-based detection. Thus, these results reveal a tangible and immediate application of our work – to design more sensitive assays for gluten detection that commonly represented urinary peptides (see discussion lines 374-380). On the other hand, we have transparently discussed that other sequences display high interindividual variability, and that further validation with targeted methods and larger cohorts will be needed to fully understand what these peptides tell us about celiac disease status and immunopathology (discussion lines 392-395). We emphasize that we have carefully and fairly discussed the implications of findings in terms of the observed variability. Overall, this manuscript resolves the decades-long question of what specific wheat peptides are formed by *in vivo* digestion, and dissemination of our method and the peptide sequences themselves sets the stage for many lines of follow up studies, as pointed out by the other three reviewers.

The other comments were all properly addressed.

We are pleased that the reviewer found that all our other responses properly addressed their concerns.

Reviewer comments, third round review:

Reviewer #1 (Remarks to the Author):

My concerns were generally not fully addressed. Specifically, (indirect) claims of superiority, comprehensiveness etc., of the protocol are still present in the paper (e.g. ... dramatically improve our ability to measure all urinary peptides,.....is less time- and labor-intensive than prior approaches, Given its high specificity (Fig. 3), our workflow should be readily applicable to other studies requiring analysis of the urinary peptidome.) and are in my eyes not justified, especially since apparently the protocol that seems to be used by the other groups working in this field has not been applied, and the results show essentially no correlation to the data reported by others on the urinary peptides.

Based on the extremely low comparability to data from other groups, it seems that the authors report on a protocol that likely enables the enrichment of a specific subpopulation of peptides, but the many of the abundant peptides present in urine (e.g. all the collagen peptides) seem to be lost during sample preparation.

The authors claim "Although here we focused on identification of wheat derived peptides, we also identified over 30,000 human peptides (Supplementary Datasets 1-7), which is to our knowledge, is the largest collection of urinary peptides sequenced by LC-MS/MS to date." However, there are issues: most of the highly abundant peptides consistently reported by others are absent, including essentially all collagen derived peptides. There are other issues of concern, e.g. some peptides are reported to be found only with oxidized methionine, which is at least highly unlikely. Peptides containing the oxidized methionine are generally found at lower abundance than the non oxidized form, while here the non oxidized form is not detected at all.

Overall, my initial concerns remain: this is a very preliminary study with multiple issues, not in good agreement with the current literature and the validity of the reported findings is shaky, supported by very limited evidence.

My concerns were generally not fully addressed. Specifically, (indirect) claims of superiority, comprehensiveness etc., of the protocol are still present in the paper (e.g. ... dramatically improve our ability to measure all urinary peptides,.....is less time- and labor-intensive than prior approaches, Given its high specificity (Fig. 3), our workflow should be readily applicable to other studies requiring analysis of the urinary peptidome.) and are in my eyes not justified, especially since apparently the protocol that seems to be used by the other groups working in this field has not been applied, and the results show essentially no correlation to the data reported by others on the urinary peptides.

Recognizing the reviewer's experience in urinary peptidomics, we have revised the language of our paper to tone down direct and indirect claims of superiority. In discussing the utility of our method, we now more succinctly state our claims and carefully verified that these claims are supported by our specific data along with appropriate literature citations. Our changes are pasted below for convenience:

Discussion lines 307-310:

Previous text: Therefore, we developed an extraction technique to remove these interfering compounds (**Supplementary Fig. 3**), and dramatically improve our ability to measure all urinary peptides, including those derived from wheat.

Revised text: Therefore, we developed an extraction technique to remove these interfering compounds. This allowed us to achieve our main goal of wheat peptide identification, while also improving our ability to measure urinary peptides originating from other endogenous and dietary sources (**Supplementary Fig. 3**).

Discussion lines 310-316:

Previous text: Our workflow (**Fig. 1**) is compatible with standard reversed-phase LC-MS/MS instrumentation available in most proteomics laboratories⁴². It is less time- and labor-intensive than prior approaches, while also allowing the identification of 2-to-10-fold more endogenous human peptides from typical 1-10 mL urine samples^{32,33,43,44}. This methodological advance allowed us to undertake a comparative analysis of the wheat-derived urinary peptidomes of patients with CeD and healthy controls.

Revised text: Our workflow (**Fig. 1**) is compatible with standard reversed-phase LC-MS/MS instrumentation available in most proteomics laboratories⁴². Compared to published approaches employing solid phase extraction techniques, our method identifies approximately 2 to 10 times more endogenous human peptides from typical 1-10 mL urine samples^{32,33,43,44}. Moreover, our sample preparation technique requires only ~6 hours, facilitating sufficient throughput for us to undertake comparative analysis of the wheat-derived urinary peptidomes of patients with CeD and healthy controls.

(Note: We have chosen references 32, 33, 43, 44 as the key references to cite for urinary peptidomics because these publications identified the greatest number of urinary peptides prior to our work.)

Discussion lines 316-317:

Previous text: Given its high specificity (**Fig. 3**), our workflow should be readily applicable to other studies requiring analysis of the urinary peptidome.

Revised text: Given its high specificity (**Fig. 3**), our workflow has potential utility for other studies requiring analysis of the urinary peptidome.

Based on the extremely low comparability to data from other groups, it seems that the authors report on a protocol that likely enables the enrichment of a specific subpopulation of peptides, but the many of the abundant peptides present in urine (e.g. all the collagen peptides) seem to be lost during sample preparation.

As we discussed extensively in the previous round of review, we could similarly argue that other protocols also only enrich specific subpopulations of peptides. We have verified that no claims appear in our manuscript which would suggest that our protocol allows enrichment of all peptides.

Also as discussed in the previous round of review, the apparent lack of endogenous collagen peptides in our datasets is not a result of loss during sample preparation. Instead, these peptides do not appear because we chose not to include hydroxyproline as a variable modification in our database searches. Inclusion of an additional variable modification of the proteome was not feasible on a reasonable timescale given our limited computational power. In our previous response focusing on the subset of data most important to our key findings, we showed that inclusion of hydroxyproline dramatically increased collagen peptide identifications. More importantly, however, while this increased our coverage of collagen peptides, we showed that no peptides were misidentified when hydroxyproline was not included as a variable modification. Moreover, as shown in our previous response letter, no significant differences in the overall number of human or wheat peptide identifications were observed when we reanalyzed this data subset with inclusion of hydroxyproline.

However, we recognize that like the reviewer, readers may find it useful to understand why we report a limited number of collagen peptides compared to prior studies. Therefore, in Discussion Lines 320-326, we now explicitly justify our choice to exclude hydroxyproline from our searches:

Parenthetically, we note that previous surveys of the urine peptidome found that hydroxyproline-modified collagen peptides were among the most abundant^{32,33,43,44}. However, here we did not allow hydroxyproline as a variable modification in our database searches, as this modification was not relevant to our overriding goal of detecting biologically relevant, wheat-derived peptides. Therefore, we have deposited our raw data in the PRIDE database to facilitate identification of additional peptides (e.g., by including other variable modifications or using alternative search engines).

The authors claim "Although here we focused on identification of wheat derived peptides, we also identified over 30,000 human peptides (Supplementary Datasets 1-7), which is to our knowledge, is the largest collection of urinary peptides sequenced by LC-MS/MS to date." However, there are issues: most of the highly abundant peptides consistently reported by others are absent, including essentially all collagen derived peptides. There are other issues of concern, e.g. some peptides are reported to be found only with oxidized methionine, which is at least highly unlikely. Peptides containing the oxidized methionine are generally found at lower abundance than the non oxidized form, while here the non oxidized form is not detected at all. Overall, my initial concerns remain: this is a very preliminary study with multiple issues, not in good agreement with the current literature and the validity of the reported findings is shaky, supported by very limited evidence.

We have addressed the reviewer's comment regarding collagen peptides in the revised Discussion (see prior paragraph). Additionally, in the prior round of review, we showed, using data cited by the reviewer, that ~68% of peptides consistently detected by other groups cited by the reviewer are indeed also present in our data. We have made no claims regarding the abundance of peptides with oxidized methionine in our manuscript. Additionally, we surveyed the literature, but we could find no claims to support the reviewers' assertion that the ratio of unoxidized to oxidized peptides is expected to be high in urine samples, which are stored in the bladder for hours prior to voiding and thus may undergo a high degree of spontaneous oxidation.